

# The sensitivity of satellite microwave observations to liquid water in the Antarctic snowpack

Ghislain Picard[1, 2], Marion Leduc-Leballeur[3], Alison F. Banwell[4], Ludovic Brucker[5], and
Giovanni Macelloni[3]

[1]Univ. Grenoble Alpes, CNRS, Institut des Géosciences de l'Environnement (IGE), UMR 5001, Grenoble, France
[2]Geological Survey of Denmark and Greenland (GEUS), 1350 Copenhagen, Denmark
[3]Institute of Applied Physics "Nello Carrara", Sesto Fiorentino, Italy
[4]Cooperative Institute for Research in Environmental Sciences (CIRES), University of Colorado Boulder, Boulder, USA
[5]Center for Satellite Applications and Research NOAA/NESDIS and the U.S. National Ice Center, College Park MD, USA

**Correspondence:** Ghislain Picard (ghislain.picard@univ-grenoble-alpes.fr)

**Abstract.** Surface melting on the Antarctic Ice Sheet has been monitored by satellite microwave radiometery for over 40 years. Despite this long perspective, our understanding of the microwave emission from wet snow is still limited, preventing the full exploitation of these observations to study supraglacial hydrology. Using the Snow Microwave Radiative Transfer (SMRT) model, this study investigates the sensitivity of microwave brightness temperature to snow liquid water content at frequencies from 1.4 to 37 GHz. We first determine the snowpack properties for 8 selected coastal sites by retrieving profiles of density, grain size and ice layers from microwave observations when the snowpack is dry during winter time. Second, a series of brightness temperature simulations is run with added water. The results show that: i) a small amount of liquid water ($\approx 0.5\,\mathrm{kg\,m^{-2}}$) can be detected, but the actual amount can not be retrieved in the full range of possible water contents, ii) the detection of a buried wet layer is possible up to a maximum 1 to 6 m depth depending on the frequency (6–37 GHz) and on the site, iii) surface ponds and water-saturated areas may prevent melt detection, but the current coverage of these water bodies in the large satellite field of view is presently too small in Antarctica to have noticeable effects, iv) at 1.4 GHz, while the simulations are less reliable, we found a weaker sensitivity to liquid water and the maximal depth of detection is relatively shallow (<10 m) compared to the typical radiation penetration depth in dry firn ($\approx 1000\,\mathrm{m}$) at this low frequency. These numerical results pave the way for the development of improved multi-frequency algorithms to detect melt intensity and depth in the Antarctic snowpack.

## 1 Introduction

Surface melting is frequent in summer in the coastal margins of the Antarctic continent and in particular on the low-elevation glacier termini; the ice shelves (Zwally and Fiegles, 1994). Monitoring melt is important not only as a climatic indicator, but also because of the potential influence of meltwater for the ice sheet's future fate. In the relatively cool conditions of the Antarctic, most of the meltwater usually refreezes in the firn, close to where it is produced. Runoff to the ocean is a small component of the meltwater cycle (Bell et al., 2017) and has a small contribution to sea-level rise, unlike on the Greenland



Ice Sheet (Agosta et al., 2019). However, through meltwater refreezing in place, an indirect and complex chain of processes leading to a potential significant mass loss has been evidenced (Scambos, 2004). Meltwater contributes to firn warming and air depletion which slowly weaken ice-shelf structure over decades (Munneke et al., 2014; Lenaerts et al., 2016; Bell et al., 2018).

In parallel, the small component of meltwater that does not refreeze in the firn, may form supraglacial lakes (Banwell et al., 2014; Arthur et al., 2020). The drainage of these lakes enhances deep crevasse formation by hydro-fracturing, providing one of the possible final triggers leading to shelf collapse (Banwell et al., 2013; Bell et al., 2018; Robel and Banwell, 2019; Leeson et al., 2020). Sea level rise follows from the resulting acceleration of the upstream land glaciers, against which the ice shelf previously acted as a buttressing force (Cook et al., 2005).

Surface melting occurrence is also a valuable climate indicator. The energy available for melt in Antarctica is relatively scarce, even in summer on the low-lying ice shelves, and is controlled by a variety of climatic variables (Jakobs et al., 2020). Warming of ice shelf surfaces by the air is a major driver (Torinesi et al., 2003a), but shortwave radiation and the intense downwelling longwave radiation during atmospheric rivers (Wille et al., 2019) are also important drivers. Locally, foehn effect has been shown to drive melt patterns on the Larsen C shelf at the foothills of the Graham Land mountain barrier (Luckman et al., 2014; King et al., 2017). On the Peninsula, the occasional winter melt events are usually attributed to foehn winds (Munneke et al., 2018) and atmospheric rivers (Wille et al., 2019, 2022). At the continental scale, no long term positive trends are apparent in the melt records over ice shelves (1979-2021) so far, despite local evidenced warming (Johnson et al., 2021). However change in the westerlies strength blowing around the continent has been shown to be significantly correlated with melt occurrence (Picard et al., 2007; Johnson et al., 2021). The El-Nino Southern Oscillation also imprints a subtle signature 40 on melt in the coasts of the Pacific sector (Johnson et al., 2021). For all these reasons, the ability to accurately monitor melt in Antarctica is important.

The detection of the state of surface melting using remote sensing has received a lot of attention since the inception of spaceborne microwave radiometry in 1970's (Ridley, 1993; Zwally and Fiegles, 1994; Abdalati and Steffen, 1997; Torinesi et al., 2003a; Colosio et al., 2021) and later using scatterometers and synthetic aperture radars (Kunz and Long, 2006; Johnson 45 et al., 2020). The detection is in principle relatively easy because the apparition of liquid water in snow induces a sharp changes in the microwave signal. Most detection algorithms used in Antarctica monitor the microwave radiation at a single frequency and polarization, and classify the surface as melting when it reaches a predetermined threshold. Other algorithms use a combination of channels (different frequencies and/or different polarizations) or more advanced detection techniques (Liu et al., 2006) but all are similar in essence.

Despite the abundance of work and the apparent simplicity of the detection, there are several unknowns that prevent optimal use of the satellite information: 1) The minimum amount of liquid water to enable the detection is not well known (Mote and Anderson, 1995; Tedesco et al., 2007), and this amount may vary across regions and years. Knowing this minimum amount is not so important when melt observations are used as a broad climate indicator (Picard et al., 2007; Johnson et al., 2021) but becomes critical at small spatial or temporal scales (Banwell et al., 2021) or for the evaluation of firn models which predict 55 liquid water content (Kuipers Munneke et al., 2012). 2) When the wet snowpack is overlaid by a dry snow layer – e.g. after a snowfall or due to night refreezing or when meltwater percolates at depth – the microwave observations may still reveal





the presence of underlying water, because of the microwave penetration. For this reason, it is inappropriate to interpret the liquid water detected by microwave observations as "surface melting" strictly (Torinesi et al., 2003a). Nevertheless, it is not precisely known up to which depth water can be detected. This depth is likely to change during the melt season due to snow

metamorphism (i.e., grain coarsening, densification, formation of ice layers). A better estimation of this depth is required particularly in the context of multi-frequency sensor missions that could provide more advanced information on the fate of meltwater, percolation, refreezing and runoff in and on the snowpack (Leduc-Leballeur et al., 2020; Mousavi et al., 2021). 3) Most algorithms produce a "surface melting" binary indicator, but there is also a great interest in the quantification of meltwater volume and of the melt rate (Trusel et al., 2013) as predicted by surface energy budget and climate models (Kuipers Munneke

et al., 2012; Fettweis et al., 2011). However, the possibility to retrieve such advanced information from microwave observations is debated, and limitations and possible accuracy need to be assessed. 4) When the surface becomes extremely wet (i.e. a saturated water layer, running water, surface ponding), the microwave signal is affected to the point that melt detection may become impossible despite the surface being obviously wet. To our knowledge, existing products do not take this limitation into account. More specifically, the areal fraction of a pixel covered by supraglacial lakes, above which melt detection becomes

impossible, remains to be quantified.

This study works to fill the four knowledge gaps outlined above by presenting a series of sensitivity analyses using microwave radiative transfer modeling. The focus is on the multi-frequency sensors (or combination of sensors) found in operational and in-preparation radiometry missions such as SMOS (Kerr et al., 2010), SMAP (Entekhabi et al., 2010), AMSR-E, AMSR2, AMSR3 (Kasahara et al., 2020) and CIMR (Kern et al., 2020). The overarching goal is to refine the interpretation of the

datasets produced with the existing algorithms, and to pave the way to new, more advanced, melt detection algorithms. This study is particularly aimed at users who need a better and more quantitative understanding of the various microwave melt products for detailed investigations and thorough firn model evaluations.

To conduct realistic simulations with wet snow, a pre-requisite is an accurate representation of the snowpack at the onset of the melt season. Given the lack of adequate in situ snow measurements in the Antarctic coastal marginal areas, our approach

is to first retrieve the snowpack properties from the microwave observations. This is only possible when the snowpack is dry, before or after the melt season. Our retrieval method builds on ideas from Mote and Anderson (1995) and Brucker et al. (2010), but with a more advanced setup. It provides a simplified but realistic dry snowpack that can be then modified with added water in different proportions and depths to investigate the microwave sensitivity to liquid water.

The paper is structured as follows: Section 2 describes the microwave observations and the test sites. Section 3 outlines the

radiative transfer model, the approach to retrieve the snowpack properties and the conducted simulations with added water in varying amount and depth. Section 4 presents the simulation results. Section 5 addresses the implication of the sensitivity analysis for the users of melt products and for algorithm developers.



## 2 Study sites and observations

Eight Antarctic coastal sites have been selected to investigate the sensitivity of microwave data to meltwater (Table 1). Five

of them (Maudheimvida, Halvfarryggen, Larsen C, Larsen B and Roi Baudouin) are chosen due to the availability of detailed meteorological observations and melt estimates (Jakobs et al., 2020). Three other sites are located on major ice shelves and have a wide variety of conditions: Wilkins (high accumulation, presence of aquifer Montgomery et al. (2020)), Amery (low accumulation, presence of supraglacial lakes, Spergel et al. (2021)) and Shackleton (high accumulation). All the sites are on ice shelves except Maudheimvida and Halvfarryggen. With firn and ice deeper than 100 m, the influence of the underlying

substrate is only significant at L band in dry conditions and is treated appropriately in our modeling setup.

Passive microwave observations acquired by the Advanced Microwave Scanning Radiometer 2 (AMSR2) on-board Japan's Global Change Observation Mission 1st - Water "SHIZUKU" (GCOM-W1) satellite are used for the retrieval, general statistics and melt detection. Data at 6, 10, 19, and 37 GHz at vertical and horizontal polarizations are extracted from the National Snow and Ice center (NSIDC) AMSR-E/AMSR2 Unified Level 3 daily product, version 2. The product has a resolution of 25 km at

6 and 10 GHz and 12.5 km at 19 and 37 GHz. The typical brightness temperature accuracy is ±1 K. The Soil Moisture Ocean Salinity (SMOS) from the European Space Agency (ESA), the Centre National d'Études Spatiales (CNES) and the Centro para el Desarrollo Tecnológico Industrial (CDTI) are also used to provide L-band data at 1.4 GHz (Kerr et al., 2001). We use the Level 3 product (Bitar et al., 2017) downloaded from the Centre Aval de Traitement des Données SMOS (https://www.catds.fr/, last access: 14 March 2022) and selected the nearest pixel of each site from the EASE-Grid 2, which tends to be distorted in

the polar regions (around 100 km in the meridian direction and 6 km in the zonal direction). The daily brightness temperature at 50-55° viewing angle is extracted at both vertical and horizontal polarizations. The typical brightness temperature accuracy is ±2 K

In this paper, the melt is detected in both the simulations and in the observations when the brightness temperature exceeds a threshold value calculated as the June to September mean brightness temperature (referred to as "dry brightness temperature"

hereafter) + 20 K. More elaborated algorithms exist (e.g. Mote and Anderson, 1995; Torinesi et al., 2003a) but we choose the same fixed offset at all frequencies for sake of simplicity and reproducibility of the results. The value of 20 K is lower than that used by Zwally and Fiegles (1994) but corresponds to a typical value of the adaptive algorithm by Torinesi et al. (2003a). The simulation results and analysis code will be made publicly available for further experiments with other algorithms (see Data Availability Section).

## 3 Method

### 3.1 The Snow Microwave Radiative Transfer model

Brightness temperature simulations are run with the Snow Microwave Radiative Transfer (SMRT) model (Picard et al., 2018). This model is one-dimensional, and it represents the snowpack with horizontal layers. In each layer, the snow microstructure



**Table 1.** Details of the eight sites selected to investigate the sensitivity of microwave data to meltwater. AWS names are from (Jakobs et al., 2020), temperatures are from ERA5 reanalysis (Hersbach et al., 2020), melt days are from AMSR2 19 GHz H-pol channel, ice thickness is from BEDMAP2 (Fretwell et al., 2013) and Surface Mass Balance (SMB) is from RACMO (Van Wessem et al., 2014).

| Site | Coordinates | Annual (winter) temperature (°C) | Number of melt days | Ice thickness (m) | Mean annual SMB ($kg\,m^{-2}$) |
|---|---|---|---|---|---|
| Halvfarryggen (aws11) | -71.170, -6.800 | -15.8 (-22.5) | 0.0 | 886 | 625 |
| Camp Maudheimvida (aws5) | -73.100, -13.170 | -17.8 (-24.6) | 2.7 | 681 | 384 |
| Amery | -70.354, 70.948 | -20.3 (-27.4) | 29.1 | 626 | 149 |
| Shackleton | -66.123, 98.395 | -14.4 (-19.2) | 42.4 | 200 | 762 |
| Roi Baudouin (aws19) | -70.950, 26.270 | -13.7 (-19.3) | 44.1 | 376 | 294 |
| Larsen C South (aws15) | -67.570, -62.150 | -12.7 (-19.7) | 59.7 | 292 | 381 |
| Larsen B (aws17) | -65.930, -61.850 | -11.5 (-18.1) | 78.0 | 206 | 282 |
| Wilkins | -70.710, -71.940 | -9.4 (-15.5) | 105.3 | 150 | 665 |

representation, temperature and liquid water content must be prescribed. Here, we selected the exponential microstructure
representation which imposes to provide in addition the density and the correlation length for each layer (Sandells et al., 2021).

Once the snowpack is set, the model computes scattering in every layer using one of the theories available in SMRT (Picard et al., 2018). Here we selected the symmetrized strong contrast expansion, recently introduced in Torquato and Kim (2021) for general random porous media and applied to snow in Picard et al. (2022). This theory has the advantage of treating the ice and air components of the snow microstructure in a symmetrical way, so that the scattering function is continuous over the
whole range of ice fractions (0-1). Before the use of this method, these functions were discontinuous when the ice components become preponderant with respect to air in the firn (i.e. around a density of $917/2 = 457\,kg\,m^{-3}$). In a last step, SMRT solves the multi-layer radiative transfer equation using the discrete ordinate method (Picard et al., 2018). Atmospheric absorption and emission are neglected here due to the low frequencies and the relatively cold and dry Antarctic atmosphere. The output is the brightness temperature of the snowpack at two polarizations (vertical and horizontal) and at 55° incidence angle (AMSR2
specifications), close to the Brewster angle. Most simulations are run for a single point, but some are run at multiple points to account for the heterogeneity within large pixels. The resulting brightness temperature over such a pixel is the average of the results at all points.

## 3.2 Wet snow permittivity

The main effect of the liquid water in snow with respect to microwave is to increase both the real and imaginary parts of the
complex effective permittivity. Hence, estimating the effective permittivity of the water, ice and air mixture is an important step to investigate the sensitivity of microwaves to the water content, and especially to determine the minimal detectable amount





of water. However, no perfect permittivity formulation exists despite tremendous efforts in the 1970's and 1980's (Tiuri and Schultz, 1980; Sihvola et al., 1985; Hallikainen et al., 1986; Mätzler, 1987, and references therein).

In a nutshell, two main regimes of wetness must be distinguished (Colbeck, 1980) depending on the low or high volumetric
water content. In the pendular regime, the volumetric water content is low (<3–7% of the total snow volume) and the water appears as isolated inclusions in the pore space (i.e. between the ice grains), usually at the joints and in the necks, where the surface tension energy is minimal. A possible representation in this regime is to consider isolated water inclusions mixed in a dry snow background. In this case, Maxwell-Garnet (MG) mixing formula applies (Sihvola, 1999). However, a delicate choice remains for the shape of these inclusions that controls the depolarization factor in this formula (Colbeck, 1980; Mätzler, 1987).
In the funicular (> 3–7%) regime, water forms a continuous shell around the ice and the air pores become isolated (a non continuous phase). By assuming spherical ice grains coated by a thin water shell, the MG mixing formula applies if the water is the background and ice is the inclusions (Chopra and Reddy, 1986). This result is counterintuitive since the water is usually in minority, but it can be understood by the fact that the electromagnetic waves mainly interact with the outer part of the grains, that is with the thin water shell. This model coincidentally has the advantage to well describe saturated snow, as water actually
occupies the background, ice grains are isolated and air is virtually absent in this case. For testing, we also consider the MG mixture of water droplets in an ice background (called here water pocket model). In both cases, to account for the air, the Polder and van Santen formula (Polder and van Santen, 1946) can be used to mix the air and one of these "wet ice" mixtures.

A third regime described by Colbeck (1980) for dense snow (density >550 kg m$^{-3}$) is not considered here.

Countless other mixing strategies and empirical fits are possible, resulting in significantly different permittivity formulations.
A number of them have been added in SMRT for this study. For a few of these formulations, Fig. 1 shows the real and imaginary parts of the permittivity as a function of the liquid water content ranging from zero (dry snow) up to the saturation point (no air) for the various sensitivity tests. All formulations agree on the fact that the permittivity (for both real and imaginary parts) increases with the water content due to the significantly higher water permittivity with respect to that of ice and air. However, differences between the formulations are large, for both the real and imaginary parts.
The formulations shown with circles in Fig. 1 are based on dielectric measurements of wet snow while the others are solely based on theoretically mixing ice, air and water with spherical inclusions. The formulation for the pendular regime by Colbeck (1980) is designed for small amounts. Likewise, the formulations by Wiesmann and Mätzler (1999) and by Hallikainen et al. (1986) (used here with the version updated by Fawwaz Ulaby (2015)) respond quasi-linearly to the water content, because they were fitted with experiments including small water contents only. They are likely invalid for high contents. MEMLS v3
(Mätzler and Wiesmann, 2007) extends the formulation by Wiesmann and Mätzler (1999) for high contents. A first group of formulations features similar behavior for small water contents, it comprises Colbeck (1980) pendular, Wiesmann and Mätzler (1999), MEMLS v3 and the coated spheres. At the other end of the range, near the saturation point, the coated spheres, the two-step mixing (ice+water)+air, and both Colbeck (1980) models converge towards similar values ($\approx 14 + j16$). The water pocket model is an outlier, which is expected given that melting snow is unlikely to be made of water inclusions in the ice
crystals.

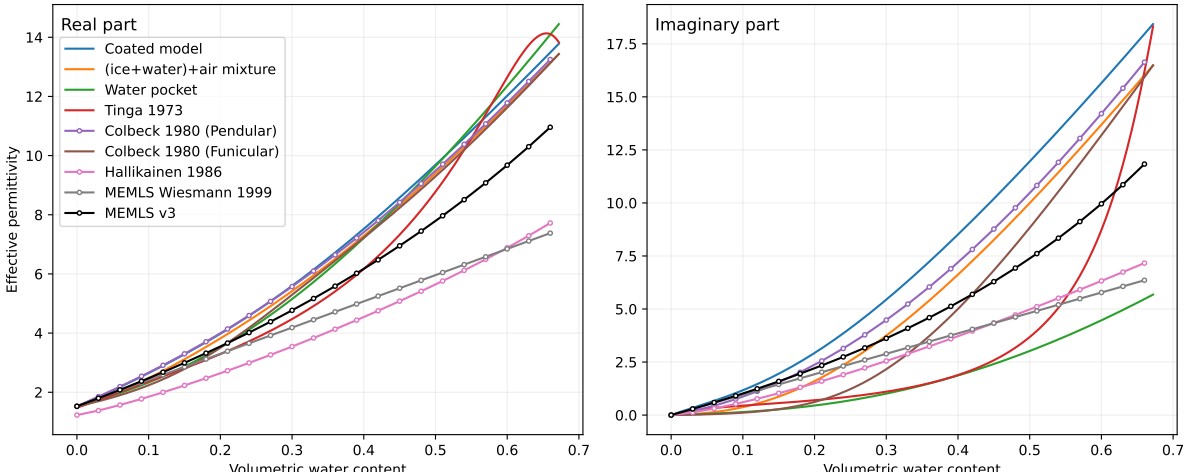

**Figure 1.** Real (left) and imaginary (right) parts of wet snow effective permittivity at $19\,\mathrm{GHz}$, calculated with different formulations for increasing quantities of water filling the pore space of snow with an initial density of $300\,\mathrm{kg\,m^{-3}}$ up to the saturation point (i.e. until no air is left in the pores).

The fundamental reason for these large differences and lack of consensus is the extreme sensitivity of the effective permittivity to the depolarization ratio, which itself depends on the shape of the inclusions. Water inclusions in the pendular regime can be very elongated depending on the ice matrix. Until new permittivity and microstructure measurements are made, new developments are unlikely. A pragmatic choice between these formulations remains the only option.

In this study, we selected the MEMLS v3 formulation for the reference simulations for the following reasons: 1) It is based on actual measurements, and 2) it behaves close to the coated spheres model and ice inclusions in water in the high water amount domain despite not being designed for that. We also considered the formulation by Hallikainen et al. (1986) and show that it can not reproduce melting snow observations.

### 3.3   Retrieval of the dry snowpack properties

The effective snowpack properties are retrieved at all test sites independently, before and just after the melt season. We call these two periods 'winter' (June-September) and 'autumn' (April-May) for convenience. The retrieval is done by searching the optimal profiles of snow properties which lead to the best agreement between the microwave observations and SMRT output. We do not consider the temporal variations during the periods, only the averaged observations are used as input and the output is a single "average" snowpack for each period. This retrieval approach builds upon previous work for the dry snowpack on the Antarctic plateau (Brucker et al., 2010), and is also somewhat comparable to Mote and Anderson (1995) where the scattering of the snowpack is first optimized in winter to then better detect summer melt. Our approach differs mainly due to our consideration of a more complex snowpack. Specifically, here, the unknown snow properties include the correlation length, snow density and ice layer number density (given in number per meter). This triplet is able to drive the most dominant pattern



of variations in brightness temperature. Density mainly controls the absorption and scattering, grain size controls the scattering,
and the ice layer number controls the difference between the horizontal and vertical polarizations (H-pol and V-pol hereinafter).
The surface roughness is neglected after preliminary tests performed with SMRT and the IEM rough surface model (Brogioni
et al., 2010) to determine the sensitivity to this snowpack characteristic. The properties profiles are assumed to be piece-wise
linear functions with four tie points at depths of 0, 3, 8 and 20 m, chosen to cover the range of e-folding depths at 6 GHz and
higher frequencies. These tie points are the main unknowns of the optimization problem. At depths >30 m, the density is set
constant to a typical value of $912\,\mathrm{kg\,m^{-3}}$ for deep firn (Burr et al., 2019) down to the base of the ice shelf (ice thicknesses are
given in Table 1), where a saline water interface is added.

The temperature profile is prescribed for each season and site. It is approximated by superposing the seasonal temperature
cycle penetrating down to $d = 2\,\mathrm{m}$ depth (Picard et al., 2009) and the overall steady gradient within the ice shelf resulting
from the difference of surface and bottom temperature. The temperature profile is described by $T(z) = (T_s - T_a)\exp(-z/d) +$
$T_a z/H + T_w(H-z)/H$ where $T_a$ and $T_s$ are the mean annual and mean seasonal air temperatures, $T_w = -2°C$ the ice shelf
bottom temperature, and $H$ the shelf thickness. Air temperatures are from the ERA5 reanalysis (Hersbach et al., 2020).

To compute the optimal values and uncertainties at the tie-points, we use Bayesian inference (Martin, 2022). The reason of
this choice is first because we have only 8 observations (6, 10, 19 and 37 GHz at two polarizations, as L-band data is unused
at this stage) for 12 unknown tie-points (4 depths for 3 properties). The minimization problem is under-determined, and the
set of optimal properties is not unique. The second motivation is to account for the uncertainties in the model results with
respect to the observations. We proceed as follows: each unknown is given a prior distribution and the goal is to compute
the joint posterior distribution of the unknown parameters of the model given the microwave observations. For the prior, we
consider that little is known about the snowpack in these regions, and choose uniform prior distributions with very wide
ranges (also called uninformative priors) to avoid constraining the results with incorrect assumptions. For instance the prior
tie-point densities are sampled in the ranges 200-700, 300-800, 400–910, 400-910 $\mathrm{kg\,m^{-3}}$ respectively at the four depths.
For the densities, we also add a constraint, the profiles with decreasing densities with depth are attributed a lower probability
than those with increasing densities. The likelihood is set by assuming that the observations are normally distributed with
zero mean (no bias between model and observation) and an unknown standard deviation $\sigma$ to be determined by Bayesian
inference at the same time as the 12 other unknowns. This standard deviation accounts for the observation uncertainties and
representativeness error of the model (i.e. the simplification and the scale mismatch). After the priors and likelihood are setup,
we compute the posterior with the Differential Evolution Adaptive Metropolis (DREAM) algorithm. This algorithm is an
efficient Markov Chain Monte-Carlo (MCMC) method (ter Braak and Vrugt, 2008; Laloy and Vrugt, 2012; Shockley et al.,
2017). 16 chains (recommended range $d/2 - 2d$, with $d$ the number of unknowns) are run in parallel over 2000 iterations and
the first 50% samples are discarded ("burn-in"). The output is an ensemble of 16000 tie-point sets following the posterior
probability. However, the MCMC methods tend to produce auto-correlated samples and in our case, the effective sample size
(Martin, 2022) is estimated ≈100. To assess the MCMC convergence, we calculated the potential scale reduction factor ($\hat{R}$)
(Gelman and Rubin, 1992) as provided by the ArviZ software (Kumar et al., 2019). Overall we obtain $\hat{R}$ values lower than
1.2 for all parameters and all sites (in the burned-in ensemble) which is acceptable. For each site, the most probable posterior





set (Maximum A Posteriori, MAP) is selected in the ensemble and used in the following sensitivity analyses. We also conduct
some simulations with a large number of samples to investigate the impact of retrieval uncertainty on the sensitivity analysis.

In summary, after the optimization phase, we have, for each site and each season, many parametrized snowpacks to simulate
the microwave observations and one of them is the optimal. While it is tempting to analyze the retrieved properties of those
snowpacks, it is important to recall that the problem is under-determined and the snowpack representation simplified. Many
equifinal sets of parameters gives similar brightness temperatures, despite they may depict quite different snowpacks from one
another, and from the real snowpack as well (Beven and Binley, 1992).

### 3.4 Wet snow simulations

To investigate the sensitivity of satellite microwave observations to liquid water, we ran simulations with the optimal dry
snowpacks (for the winter season, unless otherwise stated) to which water was added in various quantities and at various
depths. Water was always added by filling the air pores, which means that the ice mass (i.e. the dry snow density) is kept
constant. In addition the seasonal temperature $T_s$ is set to 273 K to generate the temperature profile.

We consider the following numerical experiments: Experiment 0: dry snowpack, Experiment 1: increasing water quantity
in the superficial 10 cm-thick layer, Experiment 2: increasing water quantity in the superficial 1, 2, 5, 10, 50 cm-thick layer,
Experiment 3: increasing depth of a wet snow layer with a fixed volumetric water content and thickness ($1.5 \, \mathrm{kg \, m^{-2}}$ and
10 cm), Experiment 4: mixed pixel with a varying proportion of wet snowpack and supraglacial lakes.

### 240   4   Results

#### 4.1   The dry snowpack – Experiment 0

Figure 2 shows the observed (triangles) and simulated (circles) winter brightness temperatures for each site at 4 AMSR2
frequencies and at SMOS frequency in V-pol and H-pol. The sites are sorted by increasing number of annual melt days.

The brightness temperature varies by a much larger extent between the sites, 190 − 240 K at 37 GHz and 150 − 235 K at
245 6 GHz, than the variations in air temperature during winter time (254 − 264 K), which indicates a dominant control by the
emissivity, that is itself determined by the snowpack structure. This indicates significantly different snowpacks from site to
site.

The general trend is decreasing brightness temperatures with increasing numbers of melt days, which is particularly clear
at the highest frequencies (19 and 37 GHz). Amery, despite a moderate number of melt days, stands out with the lowest
brightness temperature among all sites. It may be related to its distinctively low SMB. The H-pol follows in general the V-pol
variations at the highest frequency, implying a fairly constant H/V ratio over the sites (0.86 − 0.90). In contrast, the H-pol at
6 GHz presents a strong decreasing trend with the number of melt days, which translates into a decreasing H/V ratio (from 0.9
to 0.7). Intermediate frequencies have an intermediate behavior. The 1.4 GHz features large variations unrelated to the other





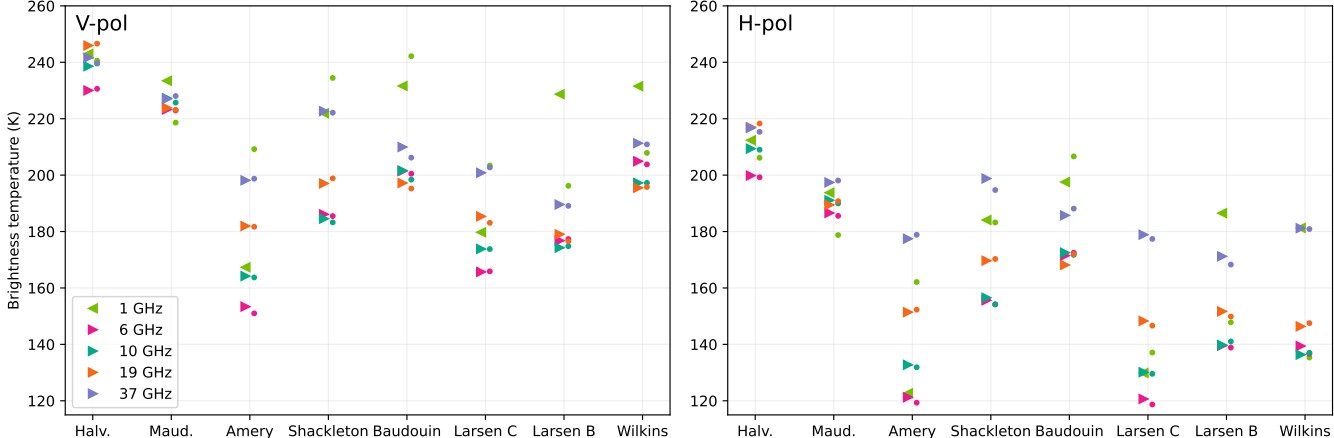

**Figure 2.** Observed (triangles) and modeled (dots) brightness temperatures at vertical polarization (a) and horizontal polarization (b) for SMOS and AMSR2 frequencies. The sites are sorted by increasing number of melt days.

frequencies. For instance, at H-pol, the brightness temperature is low and close to that at 6 GHz on Amery and Larsen C,

whereas it is much higher and close to that of 37 GHz in the other sites. The reason for this is not clear.

Overall, V-pol observations suggest that scattering is increasing with the number of melting days, probably because scattering is driven by the snow grain size, and grain coarsening in the presence of water is much faster than in dry conditions (i.e. wet snow metamorphism (Colbeck, 1982)). It must also be taken into account that fresh snowfalls counterbalance this coarsening by renewing snow at the surface with small grains. This may explain why high-accumulation areas (Shackleton and Wilkins)

feature slightly higher brightness temperatures despite a large number of melting days, and conversely why Amery has low brightness temperatures.

The H-pol observations are controlled by snow scattering and absorption exactly as V-pol, but in addition, they are sensitive to the surface density and the vertical density fluctuations in the snowpack (layering). The ice layers thus play an important role at H-pol due to high dielectric contrast between snow and ice in the upper part of the firn (Montpetit et al., 2013). At the highest

frequency (37 GHz), the microwave e-folding depth is only about one meter (e.g. 1.3 m for Halvfarryggen, and 0.75 m for Roi Baudouin). Only a limited number of layers are crossed by the upwelling radiation, and the H-pol signal is thereby close to that at V-pol, implying a high polarization ratio. At the lowest frequencies, the e-folding depth is much larger (e.g. 11.5 m for Halvfarryggen, and 5.2 m for Roi Baudouin at 6 GHz). The H-pol signal is lower than the V-pol signal due to the cumulative number of layers crossed by the upwelling radiation emitted in the snowpack at depth. The H-pol brightness temperatures and

the H/V ratios are therefore generally low at low frequencies.

The simulations with the optimal parameters for each site produce a good match with the observations at AMSR2 frequencies (dots in Fig. 2). The Root Mean Square Error (RMSE) is low; 1.3, 1.3, 1.7, 2.0 K at 6, 10, 19 and 37 GHz respectively,





accounting for both polarizations. These errors are comparable to the standard deviation $\sigma$ estimated by the algorithm (all-site posterior mean of 2.5 K).

In contrast, the results for L band (1.4 GHz) are very different. L-band simulations shown in Fig. 2 were obtained with a modified grain size profile because the simulations with the optimal parameters (not shown) present very large errors (33 K RMSE considering all sites and both polarizations). This may be partly explained by our choice to exclude the SMOS observations from the optimization. Nevertheless, the problem is more profound. To reproduce the SMOS observations, it is necessary to significantly increase scattering at 1.4 GHz (to simulate the low observed brightness temperature). This could be
done by increasing the grain size (we found that a factor of $\approx 2.8$ is necessary) but the collateral impact is degraded brightness temperatures at the higher frequencies. The snow grains are indeed small with respect to the wavelengths (Rayleigh scatterers), implying that their scattering spectral response is strongly increasing with the frequency ($\approx$ 4th power of the frequency). Brucker et al. (2010) discuss a similar issue for 19 and 37 GHz on the East Antarctic Plateau. In other words, the relatively low brightness temperatures observed at L band compared to those at higher frequencies are an indicator of the presence of large
scatterers in the snowpack, i.e. probably ice nodules or pipes (Jezek et al., 2018). We chose not to add such scatterers in our snowpack representation because it would require an increased number of unknown parameters that would not be compensated by the additional observations provided by SMOS, making our estimation problem even more under-determined than it is. Instead, we devised a pragmatic approach. The grain size is multiplied by 2.8 only for simulating the L-band brightness temperature (value obtained by successive tests), which leads to a reduced, but still relatively high, RMSE of 25 K. These results
are shown in Fig. 2. We hypothesize that this approach works for the purpose of this study based on the fact that absorption becomes the dominant process over scattering when water is added. However, in general, the L-band results must be taken with caution. Moreover, because of their peculiarities, these results are addressed in a dedicated section (Section 4.6) after the AMSR2 frequency results.

    The mean retrieved parameters for all the sites are shown in Fig. 3. Some general observations can be made despite the
risk of compensation between parameters (equifinality). The grains are systematically smaller near the surface and increase in size with depth, which is a consequence of the observed brightness temperature dependence on the frequency as explained in Brucker et al. (2010). This vertical distribution is expected because fresh, small-grained, snow is present in winter, overlying the wet-metamorphosed, coarse-grained, snow from the previous summer at depth. The sites with high brightness temperatures (Halvfarryggen and Maudheimvida) have distinctly smaller grains at 3 m depth, which is consistent with the lack of melt. The
density increases with depth. The ice layer number density at the top of the snowpack is around $1\,\mathrm{m}^{-1}$ and slightly increases for sites with greater number of melt days. At a depth of 20 m, this number is higher and fairly constant around $4\,\mathrm{m}^{-1}$ except in Wilkins. Such increase with depth can also be explained with an upper winter-snow layer with little stratification and a deeper snowpack containing ice layers that have been buried over time.

    The results for the autumn snowpack (results not shown) highlights a decrease of brightness temperature at 37 GHz compared
to the winter results, associated with a coarsening of the grains during the melt season (0.044 to 0.075 mm) and a slightly higher number of layers (1.3 to 1.5 mm) near the surface.

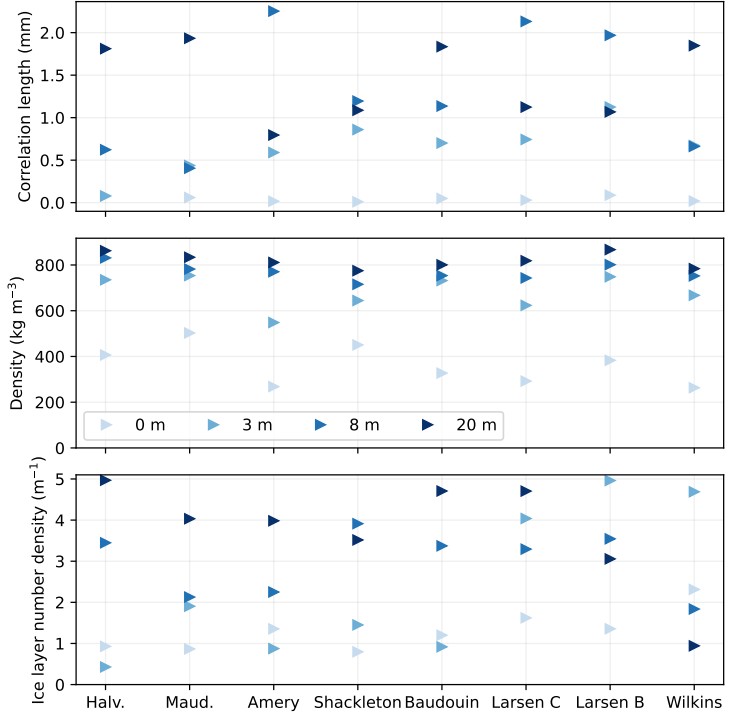

**Figure 3.** Grain size, density and ice layer number density (i.e. number of layers per meter) at depths 0, 3, 8 and 20 m for each site in winter. Each triangle is the average over the 1000 best parameter sets. Note that for Halvfarryggen grain size, the two triangles for 0 and 3 m depth overlap, and the same applies at 3 and 8 m in Wilkins.

## 4.2 Variations with the water content in the surface layer – Experiment 1

### 4.2.1 The two microwave emission regimes illustrated at one site

An increasing amount of water is added in the top 10 cm layer of the dry snowpack. The explored range is 0–20 $\text{kg m}^{-2}$, which
corresponds to 0–20% in volume in the wet layer, and about 0–50% in mass (the surface density of the best parameter set is
220 $\text{kg m}^{-3}$). Fig. 4 shows the results for Roi Baudouin, an intermediate site in terms of melt and brightness temperature.

The microwave signal is affected by the presence of liquid water in the snowpack according to two different microwave emission regimes. In the first regime, the sudden apparition of water at the surface of the ice crystal sharply increases the snow absorption because the imaginary part of the water permittivity is extremely high (Fig. 1) compared to that of ice (0.0017
at 19 GHz, Mätzler et al., 2006). As a result, the brightness temperature increases because the snowpack becomes a near-black body. At V-pol, the brightness temperature reaches 271 K, which is very close to the perfect back body (273 K). This is a consequence of the observing configuration close to the Brewster angle, the vertically polarized emitted radiation is transmitted through the surface without any loss. In reality, such high brightness temperatures are rare but can be found in 347 pixel.day



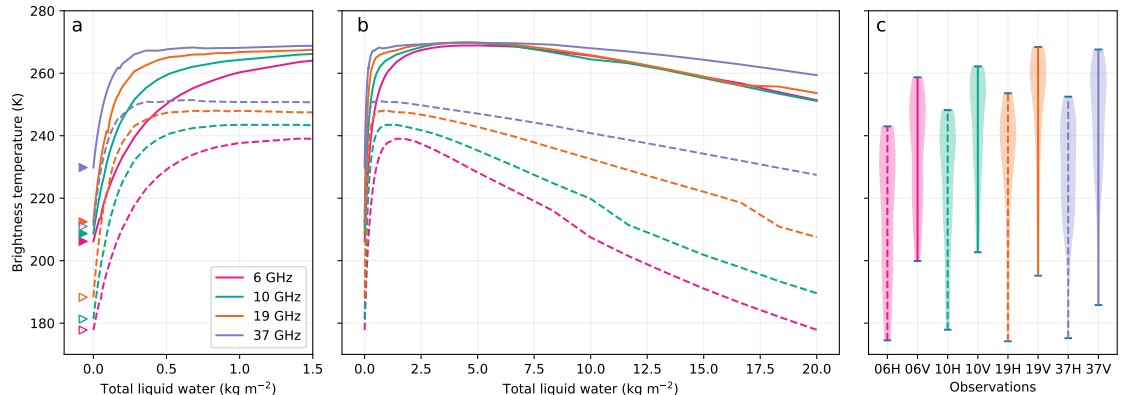

**Figure 4.** Brightness temperatures at Roi Baudouin as a function of the total liquid water content in the top $10\,\mathrm{cm}$ of the snowpack for four frequencies and at V-pol (solid curves) and H-pol (dashed curves) (a, zoomed in; and b, full range). The simulated dry brightness temperatures are marked by the triangles. The seasonal air temperature is set to 273 K which explains why the dry brightness temperatures are higher than in Fig. 2 where the winter seasonal temperature was used. Distribution of observed satellite brightness temperatures only for the days when the surface is melting (based on the melt rates computed from in situ measurements by Jakobs et al., 2020) (c).

in the 19 GHz daily 12.5 km AMSR2 records (9 summers; 2012–2021), compared to the $2.5 \times 10^6$ pixel.day with melt. In contrast, the H-pol brightness temperature only reaches 250-260 K depending on the frequency, because the surface reflects part of the snowpack emission back down. Despite this lower maximum, the amplitude of the increase is larger at H-pol than at V-pol ($\approx 70\,\mathrm{K}$ versus 55 K) which is also the case in the observations (Fig. 4c) and is the reason why most single-channel detection algorithms use the H-pol.

Another important point for the user of melt products is the water content threshold from which melt can be detected. This information is crucial for the validation of firn models for instance. We find that very small amounts of water of 0.11, 0.07, 0.05 and $0.06\,\mathrm{kg\,m^{-2}}$ can be detected (assuming a $+20\,\mathrm{K}$ H-pol increase) at 6, 10, 19, 37 GHz respectively (values for the Roi Baudouin site). The sensitivity is slightly higher at 19 and 37 GHz because of the short wavelengths. These values are similar to those reported in previous work (Tedesco and Kim, 2006) using the coated permittivity model (the strongest absorption) and using MEMLS (Figure 1 in Tedesco et al., 2007).

When the amount of water further increases, the brightness temperature reaches a maximum and then slowly decreases (Fig. 4b). In this second regime, the absorption is still very strong, but the effect of the high real part of the water permittivity becomes significant (Fig. 1). The wet surface becomes more and more reflective at H-pol (0.02 at 0% and 0.27 at 20% at 19 GHz) reducing the brightness temperature (Naderpour et al., 2017; Leduc-Leballeur et al., 2020). Then, the Brewster angle increases (from 49° at 0% to 63° at 20%) so that the V-pol brightness temperature at 55° starts decreasing as well, though in a lesser proportion than at H-pol. The same behavior is observed on wet soils and is at the basis of soil moisture retrieval algorithms for SMOS (Kerr et al., 2001). In principle, snow wetness could be retrieved in this regime using H-pol brightness temperature or H/V ratio, as investigated by Naderpour and Schwank (2018).

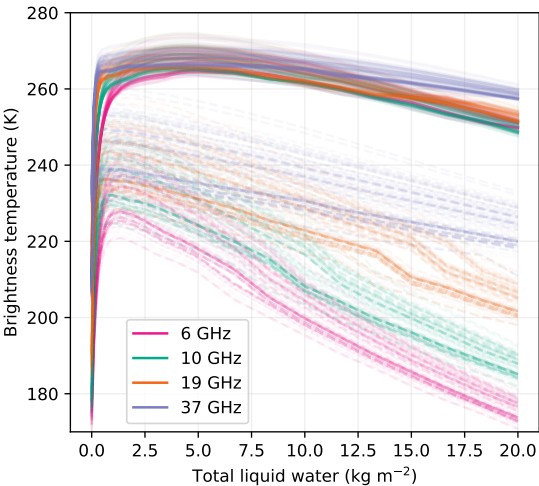

**Figure 5.** Brightness temperatures at Roi Baudouin obtained with the 1000 best (i.e. most probable) snowpacks as a function of the total liquid water content for four frequencies and two polarizations.

The transition between the first, "absorption" regime and the second, "reflective" regime takes place around 0.75 and $1.75\,\mathrm{kg\,m^{-2}}$ at 37 and 6 GHz respectively. Shi and Dozier (1995) reported a slightly higher value of $3\,\mathrm{kg\,m^{-2}}$ for radar at
C band (5.6 GHz).

Fig. 5 shows the same sensitivity analysis for a large ensemble of most probable snowpacks provided by the Bayesian inference at Roi Baudouin. It highlights the impact of the uncertainties in the retrieved parameters on the brightness temperature when water is added. In the absorption regime, the impact is relatively small, and the threshold of detection and the transition limit to the reflective regime are stable across the simulations. Conversely in the reflective regime, the variability is large,
especially at H-pol. By conducting additional simulations where all the snowpacks were set with a fixed density value at the surface (i.e. in the uppermost 10 cm layer only), we found a much weaker variability (results not shown). We conclude that the uncertainty in the surface density is the main cause of H-pol variability. This suggests that retrieving snow moisture could be improved by retrieving the surface density beforehand, a conclusion in line with the findings of Naderpour and Schwank (2018).

**4.2.2    Intra-pixel variability**

Because of the coarse resolution of the microwave radiometers ($\approx 25\,\mathrm{km}$), it is likely that the presence and quantity of liquid water in the snowpack is variable within a given pixel. Combined with the highly non-linear response shown in Fig. 4, this variability is able to affect the sensitivity analyzed in the previous section. To simulate such an heterogeneous pixel, we assume that the liquid water content follows a normal distribution, with negative values set to $0\,\mathrm{kg\,m^{-2}}$ (dry snow) and compute the
brightness temperature for 10000 liquid water values sampled from this distribution. The resulting brightness temperature over

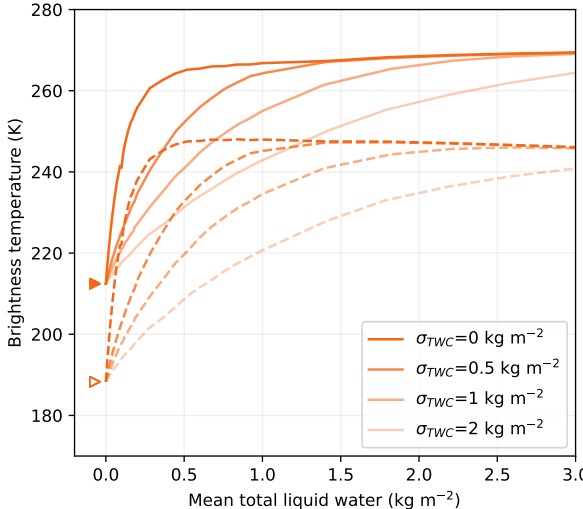

**Figure 6.** Brightness temperatures as a function of the total liquid water content at $19\,\mathrm{GHz}$ at V-pol (solid curves) and H-pol (dashed curves) at Roi Baudouin when the liquid water content is heterogeneous. $\sigma_{TWC}$ is the standard deviation of the normal distribution used to generate local total liquid water content (negative values set to 0).

such an heterogeneous pixel is the average of all these brightness temperatures. Fig. 6 shows the resulting brightness temperature at $19\,\mathrm{GHz}$ as a function of the mean liquid water content in the heterogeneous pixel. The different curves correspond to different standard deviations of the normal distribution $\sigma_{TWC}$ (before removing the negative values). The results clearly show a change of sensitivity. The rate at which the brightness temperature increases when the snowpack becomes wet becomes

smaller and smaller as the heterogeneity increases, an expected outcome of the concavity of non-linear response shown in Fig. 4. The water content threshold from which melt can be detected is about $0.9\,\mathrm{kg\,m^{-2}}$ for a $2\,\mathrm{kg\,m^{-2}}$ standard deviation, a ten-fold increase compared to $0.08\,\mathrm{kg\,m^{-2}}$ when the pixel is assumed homogeneous.

However, little is known about the distribution of surface melt rate and of the liquid water content over a 25-km pixel in reality. It is difficult to conclude whether $\sigma_{TWC} \approx 2\,\mathrm{kg\,m^{-2}}$ is a realistic value or not. The sources of liquid water content

variability include the surface roughness, the large scale terrain topography, the wind and cloudiness, dust deposition, snow density, and snow grain size, etc. Future research should investigate the distribution of melt in more detail. Meanwhile, this result demonstrates that the uncertainty in the liquid water content heterogeneity is a major contributor to the uncertainty associated with relating brightness temperature values to liquid water content. Indeed, for a given observed brightness temperature value, many different mean liquid water contents are possible, reducing the possibility of retrieving accurate values of liquid

water content from microwave radiometry.

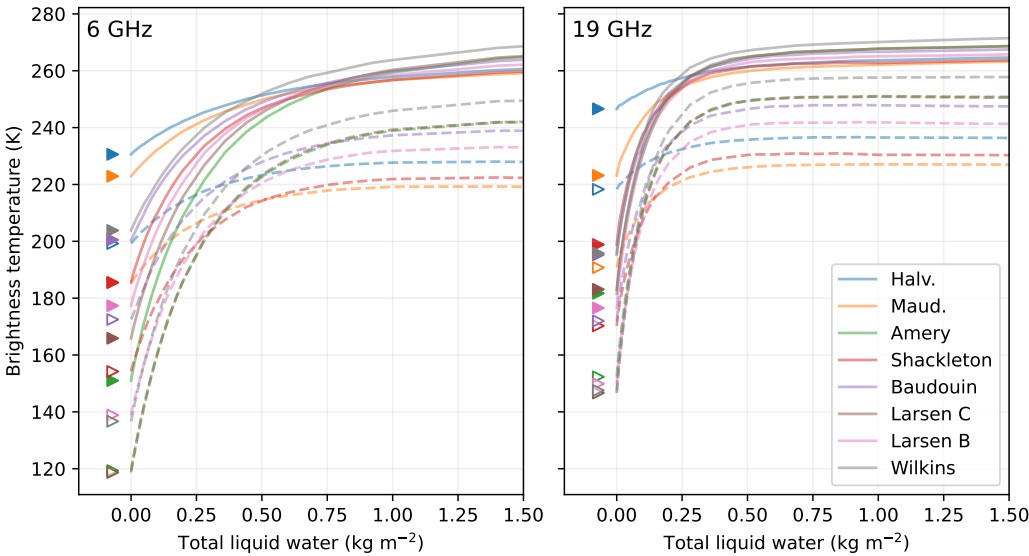

**Figure 7.** Brightness temperatures as a function of the liquid water content for all sites at V-pol (solid curves) and H-pol (dashed curves) at 6 GHz (left) and 19 GHz (right). The values of the dry brightness temperatures are marked by the triangles.

### 4.2.3 Inter-site variations

Fig. 7 shows the brightness temperatures as a function of the total liquid water content for a homogeneous pixel for the best snowpack at every site at 19 and 6 GHz. The main characteristics observed at Roi Baudouin (two regimes, threshold, sensitivity, etc.) in Section 4.2.1 appear here to be general. The variability of the dry brightness temperature is large as already mentioned in Section 4.1 and it may impact the detection sensitivity and quality. This result is well-known and all detection algorithms use adaptive techniques to cope with this variability. For instance, Zwally and Fiegles (1994) take the detection threshold to equal the averaged H-pol winter brightness temperature over 9 years plus 30 K. Torinesi et al. (2003a) take a different threshold for each year equal to the averaged H-pol winter brightness temperature plus 3 times the standard deviation H-pol winter brightness temperature. However, a potential negative consequence of such an adaptivity is a variable sensitivity to liquid water content from site to site and year to year. Our simulations indicate that the detection limit is fairly constant between 0.03 and 0.04 kg m$^{-2}$ at 19 GHz (with the simple threshold of 20 K), except for the much higher value of 0.11 for Maudheimvida and no value for Halvfarryggen which has too high dry brightness temperatures to be detected. These latter sites with the highest winter brightness temperatures – where less melt occurred in the previous years – are also those sites with the lower sensitivity and also possibly higher false-positive rate if the threshold is reduced below 20 K, in line with Tedesco (2009). This variability in the sensitivity motivated the development of a more advanced algorithm (Tedesco, 2009) where a radiative transfer model (MEMLS) estimates the brightness temperature threshold for a user-defined fixed water content depending on the snowpack characteristics.



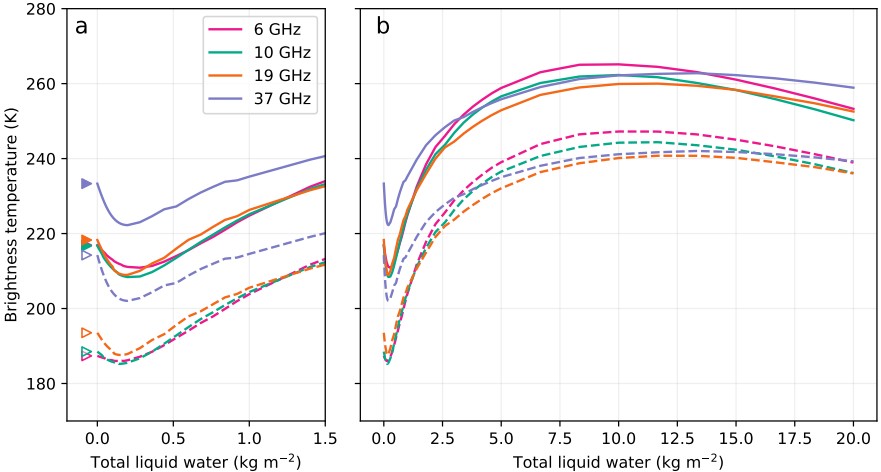

**Figure 8.** Brightness temperatures at Roi Baudouin at V-pol (solid curves) and H-pol (dashed curves) computed with the permittivity formulation from Hallikainen et al. (1986) featuring a low imaginary part as a function of the total liquid water content (a, zoomed in; and b, full range). The dry brightness temperatures are marked by the triangles.

### 4.2.4 Impact of the permittivity formulation

The results presented in the previous sections rely on the selected permittivity formulation (MEMLS v3). To illustrate the
importance of the formulation, we run simulations with the H86 formulation (Hallikainen et al., 1986; Fawwaz Ulaby, 2015) which has a relatively extreme behavior compared to the selected formulation. The results in Fig. 8 depict a very different behavior of brightness temperature for small amounts of water. A marked minimum in brightness temperature is observed around $0.5\,\mathrm{kg\,m^{-2}}$ at all frequencies and polarizations. This minimum is the signature of a "scattering regime". Indeed the H86 formulation features a particularly low imaginary part while the real part is not different from that of the other formulations.
When the snowpack is transitioning from dry to wet, the scattering is enhanced by the rapidly increasing real part (i.e. snow grains becomes more reflective) while the absorption remains weak due to the low imaginary part. The consequence is a decreasing brightness temperature with increasing water content until the absorption starts increasing significantly offsetting the scattering effect. The "scattering" regime dominates up to about $2.5\,\mathrm{kg\,m^{-2}}$ when the brightness temperature recovers to the winter value, and then further increases in the "absorption" regime as found with the MEMLS v3 permittivity formulation.
To assess the relevance of H86 formulation, we explored a large number of timeseries of AMSR2 brightness temperatures around the Antarctic and never noticed such a decrease of brightness temperature caused by weak melt events. We therefore conclude that the imaginary part of H86 formulation is certainly inadequate to model wet snow.



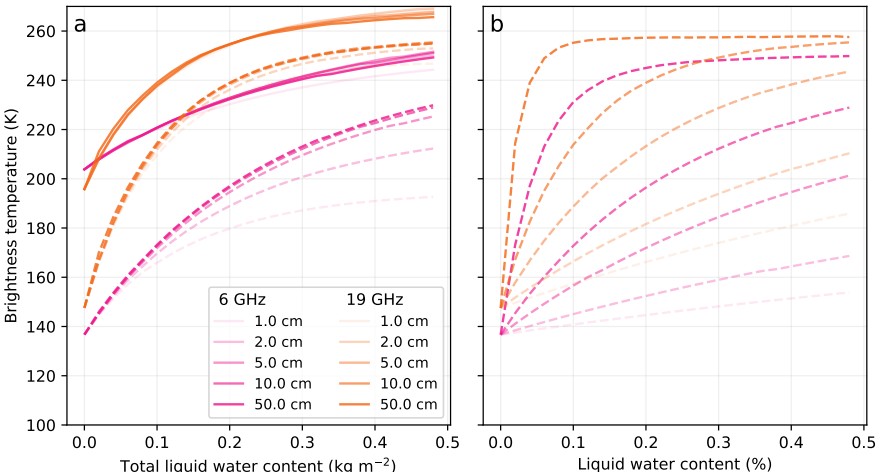

**Figure 9.** Brightness temperature as a function of the total liquid water content (a) and the volumetric water content (b) for Roi Baudouin at V-pol (solid curves) and H-pol (dashed curves) at 6 GHz (pink) and 19 GHz (orange).

### 4.3 Variations with the thickness of the wet snow layer – Experiment 2

The previous results were obtained with a fixed wet snow layer of 10 cm. Figure 9 reports a few results for different thicknesses
of wet snow (for Roi Baudouin and at 6 and 19 GHz only). We find that the thickness of the wet snow layer has little influence on the results if the total amount of liquid water is fixed when varying the thickness (Fig. 9a), and in contrast, very large variations are observed if the volumetric liquid water content (in percent of volume) is fixed (9b). This result is expected given that the brightness temperature is mainly driven by the total (column) water absorption, and that this absorption is quasi proportional to the total mass of water, for small amounts (i.e. the imaginary part of the effective permittivity is linear for small
amounts, see Fig. 1). It means that the results in the previous section generalize well if the layer thickness differs from 10 cm. Despite this, many authors usually report sensitivity in terms of volumetric water content because it is a measurable quantity and it naturally appears in metamorphism models. In this case, it is crucial to clearly report the thickness of the wet layer as well.

The case of thin wet snow layers (i.e. 1 and 2 cm) is particular. Fig. 9a depicts a smaller increase in brightness temperature
(especially at H-pol and 6 GHz) for the thinnest layers, compared to the thicker layers. We explain this specific finding by the high volumetric water content in the thin surface layer, which leads to a high permittivity constant, a high reflectivity, and in turn a low emissivity.

In conclusion, the total water content ($\mathrm{kg\,m^{-2}}$) is the main driver of brightness temperature variations with moisture, and the volumetric water content (%) is a secondary driver at H-pol. For this reason, we generally present our results in total water
content.



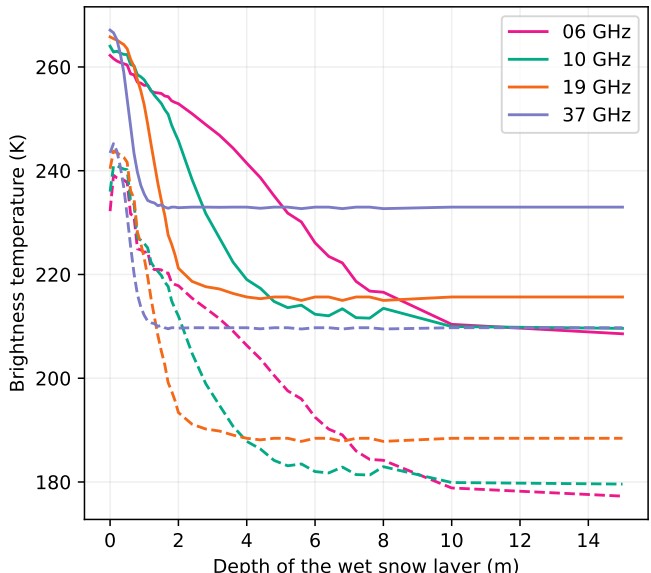

**Figure 10.** Brightness temperature of a wet snowpack as a function of the depth of a 10-cm thick wet snow layer at Roi Baudouin at V-pol (solid curves) and H-pol (dashed curves). Each curve is the average of the 100 best snowpacks to smooth the curves.

## 4.4 Influence of the depth of the wet snow layer – Experiment 3

Because of refreezing at night or snowfalls, it is frequent that a wet layer is covered by dry snow. How the brightness temperature is affected and how deep wet snow can be detected at a given frequency is an important question when investigating refreezing (Leduc-Leballeur et al., 2020). Fig. 10 shows the brightness temperatures when a 10-cm wet snow layer ($1.5\,\mathrm{kg\,m^{-2}}$)

sinks into the dry snowpack at Roi Baudouin. If we exclude a local maximum near the surface at H-pol (due to the impedance matching reducing the reflectivity when dry snow covers the wet snow layer), the brightness temperature decreases as a function of depth and reaches an asymptotic value corresponding to the dry snowpack. The rate of decrease depends on the frequency mainly. The 37 GHz signal reaches the asymptotic value above 2 m, and this depth is about 4, 8 and 10 m at 19, 10 and 6 GHz. These depths are consistent with the typical e-folding depths estimated in the Antarctic dry snowpack (Surdyk, 2002; Picard

et al., 2009) and the decreasing trend with frequency is explained by the increasing absorption and scattering coefficients with frequency. Fig. 11 illustrates this behavior with observations from the Wilkins site. This site was selected because refreezing (or percolation) of meltwater is slow and regular, leading to a clearly-visible slow relaxation of the brightness temperatures at 10 and 6 GHz (decreasing exponential shape). In contrast, 19 and 37 GHz brightness temperatures rapidly decrease as soon as the surface meltwater refreezes in March and then slowly increases from April, especially at 37 GHz, due to the accumulation

of fresh, fine-grained, snow over the summer, metamorphosed, coarse-grained snow. This area is known for the presence of aquifers (Montgomery et al., 2020; van Wessem et al., 2020)



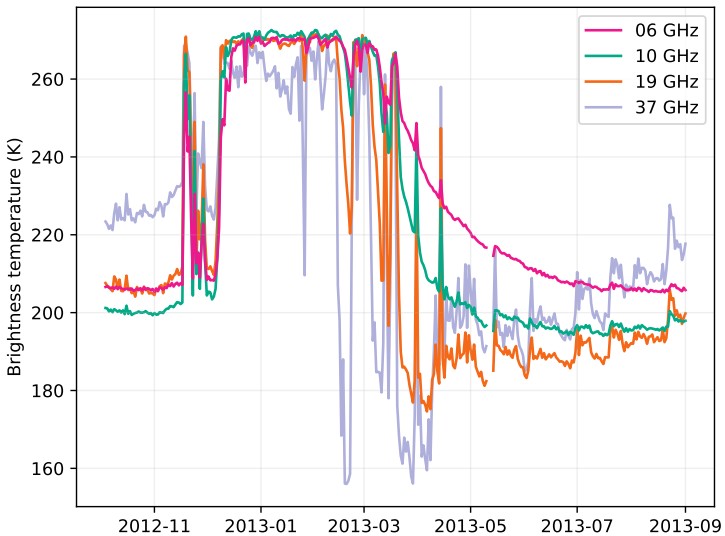

**Figure 11.** Time-series of brightness temperature at Wilkins at V-pol.

If we consider a detection threshold of 20 K above the dry mean H-pol brightness temperature, Fig. 10 indicates that wet snow can be detected up to 5.6, 2.8, 1.4, 0.6 m at Roi Baudouin for 6, 10, 19 and 37 GHz respectively. These values only partially follow the variations of e-folding depths with frequency, because the detection is also dependent on the brightness temperature amplitude between dry and wet snow. This amplitude is lower at lower frequencies, and the detection is less sensitive in general.

Interestingly these detection depths are variable from site to site and depend on the season as depicted in Fig. 12. The site dependence seems to be influenced by the dry brightness temperature (Amery and Larsen C feature the lowest winter brightness temperatures at the two/three lowest frequencies and the deepest possible detection) rather than by the e-folding depth which is expected to be reduced when the brightness temperature is lower due to coarser snow grains. At 37 GHz the pattern is less marked, probably because of the reduced penetration depth, because the snow in the topmost meter is new in winter and thereby not influenced by the previous year melt.

The maximum detection depth is generally reduced in the autumn snowpack compared to that in winter. Meltwater profoundly transforms the upper snow layers by wet metamorphism and formation of ice nodules. Larger grains and nodules lead to enhanced scattering (when the snowpack becomes dry) which reduces the e-folding depth. The lower frequencies are less affected because these processes are most active near the surface, and also because of the lesser role of scattering. This result implies that the detection at 37 GHz may be more challenging, with the necessity to adapt the threshold during the course of the melt season.

No maximum detection depth can be estimated for Maudheimvida and Halvfarryggen (the two driest selected sites) because the brightness temperature variations with the depth of the wet snow are much weaker than the threshold of 20 K and than at the

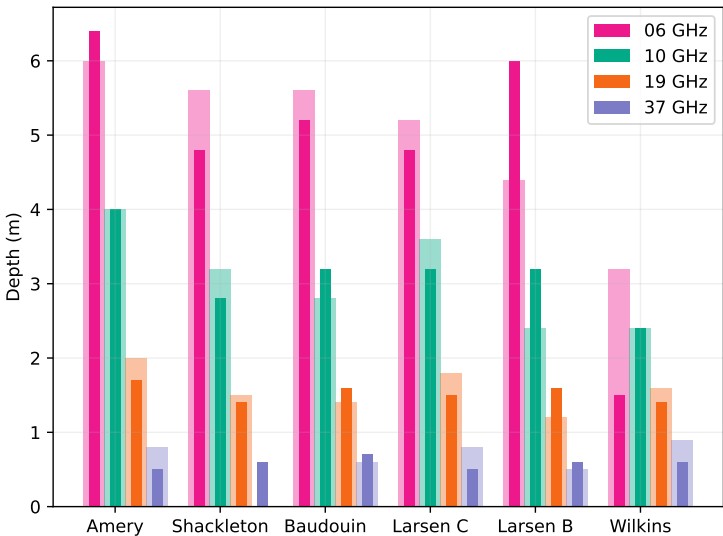

**Figure 12.** Maximum depth of detection (assuming a threshold at dry brightness temperature + 20 K) as a function of season, site and frequency. Winter and autumn are in dark and light colors respectively. Melt in the first two sites is never detected with the aforementioned criterion.

other sites (Fig. 13 for Halvfarryggen), probably because the snowpack is already close to a black body even when dry (small grains, high density according to our retrieval). With a smaller threshold, the melt could be detected in H-pol at low frequencies according to SMRT but not at high frequencies. This unexpected behavior is confirmed by the observed timeseries of brightness temperature in Fig. 13. 6 and 10 GHz features the characteristic increase of a melting snowpack in summer although the higher

frequencies follow the annual temperature cycle. The surrounding pixels (not shown) are alike, ruling out a collateral effect of the field of view increasing with decreasing frequency. The V-pol behavior differs between SMRT simulation and the time-series (different mean level and less variations in the simulation) but the unexpected frequency-dependent behavior is again observed.

### 4.5 Saturated layer and supraglacial-lake – Experiment 4

In experiment 1, we evaluated the influence of the water content for small to moderate fractions (up to $20\,\mathrm{kg\,m^{-2}}$ which is equivalent to 20% water volume) and showed a decrease of H-pol brightness temperature in the reflective regime. In the case of intense melt and multiple freeze-thaw cycles, we find that an impermeable horizon may form at or below the surface. By preventing percolation, meltwater may accumulate above this horizon, leading to high fractional volumes of water. In the most extreme case, a supraglacial lake may form (no ice crystals are present at the surface). As soon as the thickness of such a

saturated layer exceeds a quarter of the wavelength (a few millimeters depending on the frequency), the brightness temperature strongly decreases due to the high reflectively of this layer. Here we investigate the risk of misclassifying such a pixel as not

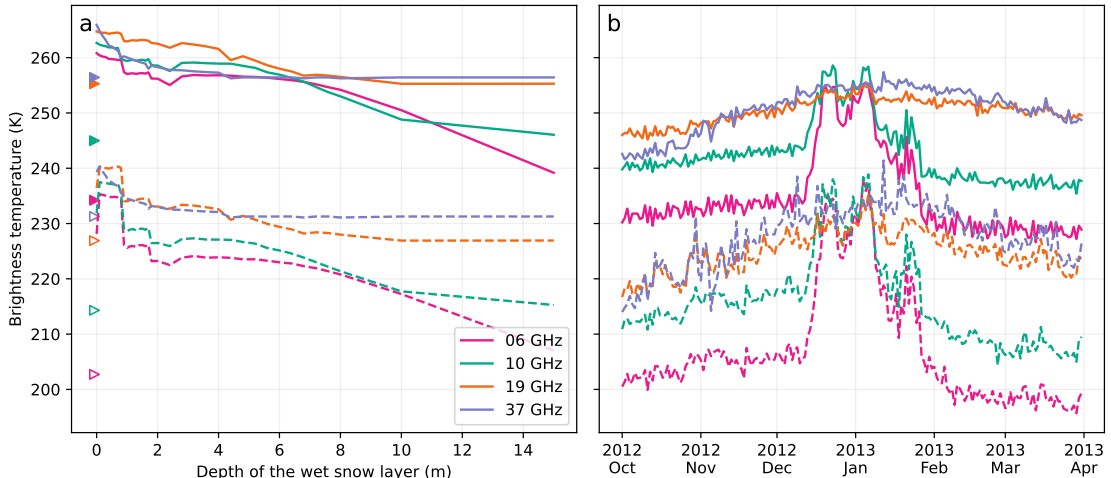

**Figure 13.** Brightness temperature as a function of the depth of the 10-cm thick wet snow layer at Halvfarryggen at V-pol (solid curves) and H-pol (dashed curves) (a). Values of dry brightness temperatures are marked by triangles. Each curve is the average of the 1000 best snowpacks. Timeseries of brightness temperature observations at Halvfarryggen (b).

melting, despite the obvious presence of meltwater. Two cases are studied: a snow layer saturated with water and supraglacial lakes.

Saturated snow is modeled as a mixture of ice scatterers in a water background. Fig. 14 shows the variations of brightness
temperature when the proportion of water increases from pure ice to pure water. When this layer is exactly at the surface (solid lines) the brightness temperature first increases (due to the increasing absorption), and then decreases to very low values (due to the increasing reflectivity of the wet interface). Browsing the AMSR2 dataset, such values much lower than the winter average have never been observed on glacial ice (it is however common on sea-ice), which questions the realism of this simulation.

Another set of simulations with the saturated layer overlaid by a thin wet snow layer was run (dashed lines in Fig. 14).
The signal is completely different because the wet snow layer (5 cm thick and total water content of $5 \, \mathrm{kg \, m^{-2}}$) is a strong absorber, so no radiation emerges from the saturated layer below and its emission is close to that of a black body, so it has a high brightness temperature. These simulations highlight the critical role of the surface conditions. If the snow is wet but not saturated in the upper snowpack (a thickness of a quarter wavelength is sufficient), the saturation effect and everything below has no influence.

Supraglacial lakes correspond to a water fractional volume of 100% in Fig. 14, characterized by a very low brightness temperatures. However, the coverage of supraglacial lakes is never complete at the scale of the microwave radiometer resolution ($\approx 12$-$50 \, \mathrm{km}$) in Antarctica; a pixel contains some fraction of ponded lake water and the remaining area is snow. To investigate this situation, which is frequent on the ice shelves, we simulate the pixel-averaged brightness temperature by linear combination of a simulation with a flat, pure-water surface and a simulation with a wet snowpack with $1.5 \, \mathrm{kg \, m^{-2}}$ total water content as
in Experiment 1. The results in Fig. 15 show a strong negative trend in brightness temperature as a function of supraglacial





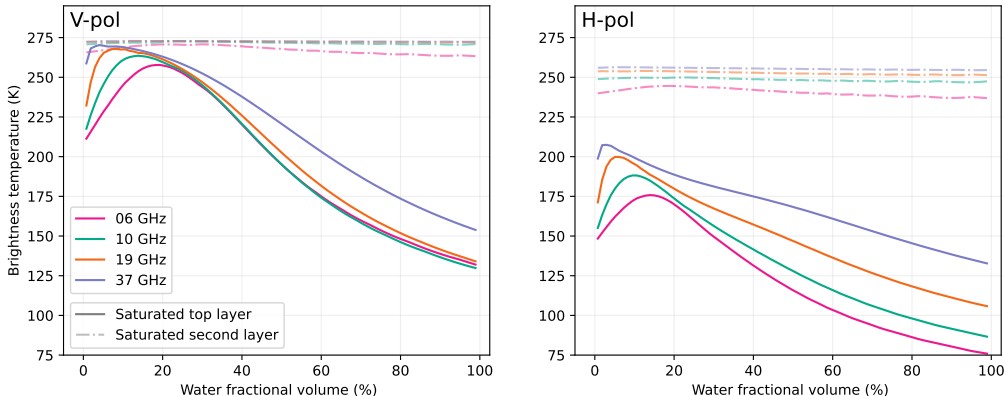

**Figure 14.** Brightness temperature of a water-saturated layer as a function of the water proportion (the remaining proportion is ice). The saturated layer is located at the top of the snowpack (solid lines) and $5\,\text{cm}$ below a wet snow layer ($5\,\text{kg}\,\text{m}^{-2}$, dashed lines).

lake fraction at both polarizations. When this trend intersects the threshold of detection (shown as a dashed horizontal line for each frequency and polarization, assuming the threshold at dry brightness temperature $+20\,\text{K}$), the mixed pixel is detected as non-melting, which is a false negative. This occurs for lake fractions larger than 42–48% at 6 GHz and 57–60% at 19 GHz, the lower and higher bounds corresponding to the H-pol and V-pol respectively. Such high areal lake coverage fractions have rarely

been observed in Antarctica (Arthur et al., 2022). For example, even before the near-complete collapse of the Larsen B Ice Shelf in 2002, which has been attributed to about 3000 rapidly draining lakes (Banwell et al., 2013; Robel and Banwell, 2019), the fractional lake coverage reached a maximum of only 10% (Banwell et al., 2014). Or in more recent melt records on shelves subject to ponding, such as the north George VI Ice Shelf where a maximum of 15% lake coverage was observed (Banwell et al., 2021), or in our observations of the eastern Roi Baudouin Ice Shelf, we have not identified a significant decrease in

brightness temperature in the AMSR2 dataset.

We can conclude that false negative melt detection due to supraglacial lake is unlikely at present time in Antarctica. In contrast, the Greenland Ice Sheet is subject to more intense surface melting in the ablation area (Bell et al., 2018). For example, a recent case of flooding on the ice sheet was reported in August 2021 (Box et al., 2022) after an atmospheric river, combined with rain, triggered a massive melt event. The decrease of brightness temperature is visible although the surface was probably

wet and remained wet for many days after this event.

### 4.6 The sensitivity of the L band to liquid water

Using the grain size adjusted snowpack, we assess the sensitivity of the L band to liquid water, first in the top layer with a similar setup as Experiment 1, except that the wet layer is $30\,\text{cm}$ thick to account for the much longer wavelength at L-band. The analysis is conducted at Roi Baudouin where the dry snowpack simulations are fairly good (Fig. 2). Fig. 16 highlights

the overall lower sensitivity compared to the higher frequencies; the increase in brightness temperature is lower (15–25 K) and

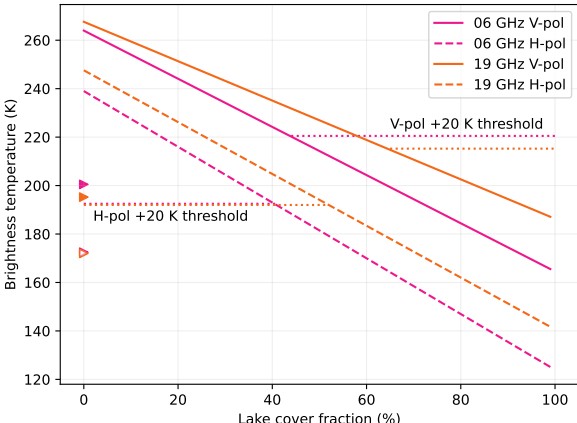

**Figure 15.** Brightness temperature of a mixed pixel as a function of lake cover fraction (the remaining area is covered by wet snowpack; 10-cm thick snow with $15\,\mathrm{kg\,m^{-2}}$ of total water content). The dry winter brightness temperatures are indicated (triangles) as well as the thresholds of melt detection (dotted horizonal lines). Note that the H-pol thresholds are very close at both frequencies, the lines almost overlap.

the saturation at V-pol is reached for a larger water amount ($>30\,\mathrm{kg\,m^{-2}}$). The H-pol reaches its maximum around $14\,\mathrm{kg\,m^{-2}}$ which is more than ten times the amount observed for the higher frequencies. This weaker sensitivity explains why lower thresholds have been used for the detection of melt with SMOS, as well as why a slightly adapted algorithm has been used (Leduc-Leballeur et al., 2020). This lower threshold does not imply larger detection errors because the brightness temperature

during the dry periods is very stable throughout the year at L band (Macelloni et al., 2016; Leduc-Leballeur et al., 2020). Hence even a change of $\approx 10\,\mathrm{K}$ is significantly larger than the noise level and the temperature-induced variations of brightness temperature.

According to the very low values of ice absorption at L band (Mätzler, 1996; Macelloni et al., 2016, 2019), a wet snow layer could contribute to the radiation emanating from the snowpack up to depths of hundreds of meters. This is in apparent

contradiction with the results in Fig. 17 showing the brightness temperature at L band with a $30\,\mathrm{cm}$ thick wet layer at increasing depths (Experiment 3). The brightness temperature decreases within a few meters and reaches an asymptotic value. This value is relatively close to the dry brightness temperature, making the melt detection difficult unless a very low threshold is used. This rapidly vanishing wet signature is due to the fact that the firn at L band at depth is only weakly scattering, so that both the wet layer and the firn at depth behave as a black body at a similar temperature, i.e. they emit the similar radiation flux. In

the case where the firn temperature is significantly lower than $273\,\mathrm{K}$ (cold site), the asymptotic value may be higher than the winter (cold) brightness temperature and the detection at greater depths may be easier.

Fig. 17 also shows a rapid increase of brightness temperature between the top wet surface ($213\,\mathrm{K}$), as in Experiment 1, and just below the surface ($235\,\mathrm{K}$) (i.e. covered by a dry snow layer). The impedance matching is the cause (i.e. the reflectivity

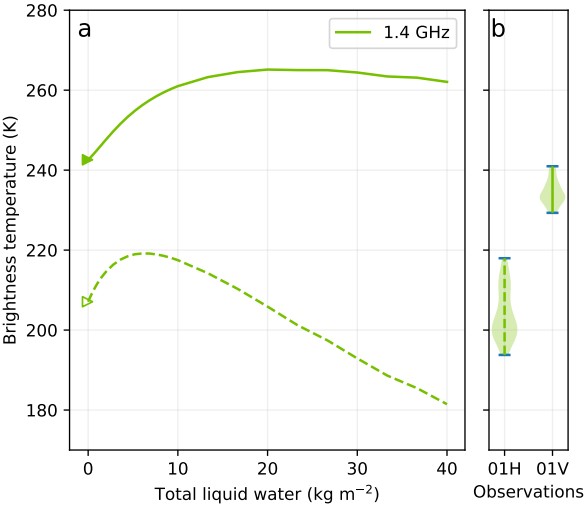

**Figure 16.** Brightness temperature at Roi Baudouin as a function of the total liquid water content in the top 30 cm of the snowpack at L band at V-pol (solid) and H-pol (dashed) (left). The simulated dry brightness temperatures are marked by the triangles. Distribution of observed brightness temperature when the surface is melting according to Jakobs et al. (2020) (right).

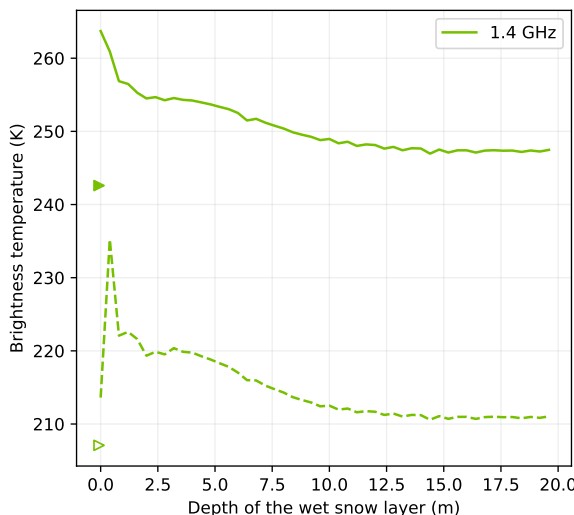

**Figure 17.** Brightness temperature at Roi Baudouin as a function of the depth of the 30-cm thick wet layer at L band at V-pol (solid) and H-pol (dashed). The simulated dry brightness temperatures are marked by the triangles.

of the surface is lower when covered by dry light snow) and the consequence is that wet snow may be easier to detect when
slightly buried than when right at the surface.





Overall, the detection of liquid water at L band is more difficult than for higher frequencies, but possible at greater depths.

## 5   Discussion and conclusions

From the perspective of the algorithm developers and the users of melt products, three important knowledge gaps can be addressed with our results and analyses.

The minimum liquid water content required for microwave radiometry to detect the presence of meltwater is generally small compared to the melt amount produced over a typical summer day in Antarctic coastal areas (Jakobs et al., 2020). Even a small melt event is likely to be detected, especially with AMSR-E or AMSR2 which have a near local noon overpass time (Picard and Fily, 2006). In contrast, the detection at L-band requires more water over a greater depth of snow. This result is confirmed in the observations (Leduc-Leballeur et al., 2020) but is subject to caution, because our simulations of the dry snowpack are

not very accurate for L-band, due to a poor representation of large scatterers in the snowpack (probably ice pipes).

For a fine analysis of the melt products, estimation of a precise minimum liquid water content for detection might be necessary. Unfortunately, this threshold depends on many variables: the algorithm parameters (e.g. the brightness temperature threshold), the wave frequency and polarization, the preexisting snowpack properties (density and grain size near the surface) and more importantly the spatial distribution of liquid water within the sensor field of view (melt heterogeneity). Some of these

variables are known a priori (e.g. sensor configuration), and their effect can be simulated by our modeling framework, but some others are subject to high uncertainties, resulting in uncertain minimum liquid water content values. To evaluate firn models against microwave-derived melt products, the usual current practice is to assume a fixed minimum water content (Fettweis et al., 2011; Kuipers Munneke et al., 2012). Instead, we suggest the exploration of a range of minimum water content between e.g. 0.2 and $1 \, \mathrm{kg \, m^{-2}}$ to test the significance of the results. For instance, a model showing a systematic positive bias for a large

range of minimum water contents is clearly positively-biased. On the other hand, a bias with changing sign when the minimum water content is varied would indicate that the model is not biased and is as good (or uncertain) as the melt product. For the algorithm developers, we suggest to provide outputs with varying parameters, as for instance in Banwell et al. (2021), where the brightness temperature threshold used to detect melt was varied in a reasonable range (i.e. from 2.5 to 3.5 $\times$ the standard deviation of the winter brightness temperature) to give a rough uncertainty estimations for the annual number of melting days.

The maximum snow depth for liquid water detection is another important parameter to enable the accurate use of microwave radiometer melt products. For instance to compute the "total" liquid water in a firn model, outputs must be evaluated from the surface to this depth. The main factor controlling this depth is the frequency, with variations from $0.1$–$0.2 \, \mathrm{m}$ at $37 \, \mathrm{GHz}$ to about $10 \, \mathrm{m}$ at $1.4 \, \mathrm{GHz}$. Taking advantage of this large depth range to provide rich information on surface refreezing and percolation is an important research avenue (Leduc-Leballeur et al., 2020; Miller et al., 2020; Colliander et al., 2022). The preexisting

snowpack properties (density, grain size and ice layers) are a secondary factor modulating this maximum depth, and since all these properties evolve rapidly in wet conditions, the depth is subject to seasonal variations. We have shown that the retrieval of these three snow properties is feasible before and after the melt season, but the wet snowpack remains opaque during the melt season, forbidding the retrieval of any information under the wet layer. Additionally, as a general rule, the maximum depth of



detection decreases slightly over the course of a melt season due to grain coarsening, densification, and ice layer formation that
occur in wet conditions.

The two regimes of variations in the brightness temperature results in non-monotonic variations as a function of liquid water content. This could be an issue for melt detection as very wet snow has a low brightness temperature (especially at H-pol), potentially below the melt detection threshold. Our results show that the required amount of water concentrated in the upper layer is large and is not typical of the Antarctic environment at the present time. While the H-pol is often favored over the
V-pol for melt detection (Zwally and Fiegles, 1994; Tedesco et al., 2006; Torinesi et al., 2003b), future algorithms could assess the benefit of using V-pol brightness temperature to avoid this problem, versus the disadvantage of the lower V-pol contrast between dry and wet conditions. Finally, in the reflective regime, for liquid water contents higher than 2–5 $\mathrm{kg\,m^{-2}}$, the H and V-pol signals can be used to retrieve liquid water content (Naderpour and Schwank, 2018; Houtz et al., 2021), but we have shown that appropriate prior knowledge of the surface density and absence of water at depth are essential.

Geographically, our study relies on specific sites in Antarctica's coastal areas, mainly ice shelves. However, most results show relatively small inter-site variations, which leads us to conclude that the benefit of this study is more general and contributes to the knowledge of the Antarctic Ice Sheet at large. Applicability to the accumulation area of the Greenland Ice Sheet (where the glacial ice is far below the microwave penetration depth) and where the number of melting days is similar, is probably acceptable. In addition, despite using AMSR2 and SMOS observations, our results also apply to any radiometric sensor
operating at frequencies of 1.4 – 37 GHz and with incidence angles of 50-60°. This includes for instance SMMR, SMAP, SSM/I, SSMIS and CIMR.

This work has provided the basis for developing new advanced melt detection algorithms, a logical future avenue of research. However, we stress that fundamental knowledge is still missing regarding the wet snow permittivity, the small scale liquid water variability, and the snowpack properties in the coastal Antarctic environment, particularly the ice shelves. Future work should
focus on filling these gaps given the growing importance of the ice-sheet hydrology as the climate warms.

*Code and data availability.* The SMRT model code, including the new permittivity formulations, is available from https://github.com/ smrt-model/smrt. The code to produce the figures will be posted as well https://github.com/smrt-model/smrt_liquid_water_paper. This repository will also include the retrieved snowpack properties and the observed and simulated brightness temperatures at the select study sites.

*Author contributions.* GP conducted the study, implemented the permittivity equations in SMRT and run the simulations. ML, GM, AFB,
and LB contributed and/or commented to the analysis and the results. All authors contributed to writing the manuscript.

*Acknowledgements.* This work is funded by the ESA Project ESA/AO/1-9570/18/I-DT – "4D Antarctica". SMRT was developed in the ESA Project 4000112698/14/NL/LvH "Microstructural Origin of Electromagnetic Signatures in Microwave Remote Sensing of Snow". Alison F. Banwell received support from the U.S. National Science Foundation (NSF) under award 1841607 to the University of Colorado Boulder.



We thank Carlo Marin for debugging the H86 permittivity formulation in SMRT. The scientific results and conclusions, as well as any views
or opinions expressed herein, are those of the authors and do not necessarily reflect those of NOAA or the U.S. Department of Commerce.

*Competing interests.*   LB is a member of the editorial board of The Cryosphere. Other authors declare no competing interests.



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
