# Peer review of "The sensitivity of satellite microwave observations to liquid water in the Antarctic snowpack"

_The Cryosphere, 2022_

## Referee Comment (RC1)

**Review for:**

"The sensitivity of satellite microwave observations to liquid water in the Antarctic snowpack"

by Ghislain Picard, Marion Leduc-Leballeur, Alison F. Banwell, Ludovic Brucker, and Giovanni Macelloni

submitted to *The Cryosphere Discussion*

Manuscript ID: TC-2022-85

**Synopsis:**

In this paper, the sensitivity of microwave brightness temperature to snow liquid water content at different frequencies from L-band to Ka-band has been investigated. Considering a recent interest in multi-frequency snow monitoring, authors have done a good job in drafting this paper. However, there are some issues that need to be addressed before this manuscript can be published in this journal. Please see below for my major comments.

**Comments to the Author:**

1. In line 66, where you mentioned that "… 4) when the surface becomes extremely wet (i.e. a saturated water layer, running water, surface ponding), the microwave signal is affected to the point that melt detection may become impossible despite the surface being obviously wet.", can you please explain why it becomes impossible? When it is very wet to the point to the point that a microwave signal is saturate (depending on the frequency of operation), a melt event for that specific season would be detected, but there may be a delay in detection (depending on frequency), or even uncertainty in the correct absolute value of liquid water content, as one frequency signal gets saturated, it cannot retrieve any further excessive wetness. And here is when multi-frequency detection becomes valuable as you mentioned. That said, please correct me if I'm thinking wrong.
2. In line 80, where you mentioned "This is only possible when the snowpack is dry, before or after the melt season.", isn't it different to use before or after melt season as a good dry snow condition? A snowpack that went through a longer winter season, will become closer to a more ideal dry snowpack; hence, a snow condition more to the winter season (further away from last day of melt season) would be a better dry snow condition. Can you please explain this a little bit here and where it's necessary in the paper?
3. In line 153, where you mentioned "A third regime described by Colbeck …", can you please add this sentence to the end of paragraph? It's too small to be a paragraph on its own.
4. In line 234, you mentioned "Water was always added by filling the air pores, which means that the ice mass (i.e. the dry snow density) is kept constant.", can you please why it has been decided to keep the density constant? Shouldn't the snow density change with snow melting, at least for the tope layers, where usually undergo metamorphism?
5. In Figure 2, please , mention in the caption the period (months of year) observed brightness temperature.

6. In line 254-255, where you mentioned "For instance, at H-pol, the brightness temperature is low and close to that at 6 GHz on Amery and Larsen C, whereas it is much 255 higher and close to that of 37GHz in the other sites. The reason for this is not clear.", first of all, it is very important when these measurements have been performed. For example, some years, some ice shelves can undergo more metamorphism, which introduces more distinct vertical layer boundaries, which in turn decreased the brightness temperature drastically, mainly in H-pol. Also, surface scattering can decrease TB if it's comparable to the wavelength, and maybe that's why L-band is close to 6GHz in Amery and Larsen C. In general, as frequency increases, the applied electric field changes polarization faster and forces the dipoles in the dielectric to change directions faster as well. Hence, dielectric loss increases. In addition, as the frequency increases, E-field changes polarization faster than the dielectric dipoles' relaxation time, which in turn decreases the displacement electric field in the material and decreases the dielectric constant. Hence, TB increases (less reflection). Perhaps, by comparing different measurement during different periods of year (or even same months but different years), you may be able to demystify why L-band behaves like this.

7. In line 256, it was claimed that V-pol suggests that scattering increases with number of melt days. Can you please explain why? This is contradictory with the statement in line 262, where you mentioned that H-pol and V-pol are controlled similarly by scattering. Your claim in line 256 could be right, but it definitely needs more proof and validation.

8. In line 266, where you mentioned H/V ratio becomes larger mainly due to the layering, can you explain this? Assuming that there is no attenuation, and no surface scattering, are we still seeing this effect? Usually, H and V observations becomes close to each other when there is enough surface scattering to make the signal to loose its coherency, then it becomes polarization-independent.

9. Please also explain why in figure 2, V-pol decreases and then increases, while H-pol monastically decreases by increasing the number of melt days.

10. Can you please compare your retrieved snow density with that of other available methods, such as "M. Mousavi, A. Colliander, J. Z. Miller and J. S. Kimball, "A Novel Approach to Map the Intensity of Surface Melting on the Antarctica Ice Sheet Using SMAP L-Band Microwave Radiometry," in *IEEE Journal of Selected Topics in Applied Earth Observations and Remote Sensing*, vol. 15, pp. 1724-1743, 2022."?

11. In line 318, what do you mean by "pixel.day"? Can you please explain this unit to avoid confusion?

12. Please specify incident angle in Figure 4, and wherever it's needed.

13. Can you please compare your result for snow wetness with other algorithms? You can convert it to liquid water column using density and thickness of wet snow layer.

14. In Figure 5, please specify in the caption that solid curve is V-pol and dashed one is for H-pol.

15. In line 404-407, it was claimed if the total liquid water content is fixed, wet snow layer thickness has little effect. First of all, can you please write an expression to relate volumetric liquid water content to the total liquid water content? If the thickness is normalized in the total liquid water content, the thickness should have zero effect with fixed total liquid water content, in other words, as thickness increases one would expect the signal loose more energy with exponential term depends on the thickness and the imaginary part of propagation constant. However, if the imaginary part is normalized by depth, and kept constant, the exponential term would be

constant. That said, I believe a bit more explanation with some equations would avoid any confusion.

16. In line 418-420, where it was claimed that total water content has more effect than volumetric water content, it is very confusing. It looks like there are two different parameters. This need to be clearly explained.

17. In line 470, where you mentioned "… saturated layer exceeds a quarter of the wavelength …", why quarter wavelength? It should depend on the penetration depth. Can you please explain where does this quarter wavelength come from?

18. In Lines 519 – 521, where you mentioned "This is in apparent contradiction with the results in Fig. 17 showing the brightness temperature at L band with a 30 cm thick wet layer at increasing depths (Experiment 3)." If there is a very wet layer, such that the thickness is greater than the penetration depth of the frequency of operation, it becomes a very reflective layer, and drops the TB, no matter what depth this layer has been resided. S, it is very confusing that you are claiming it is in apparent contradiction with (Mätzler, 1996; Macelloni et al., 2016, 2019) for L-band. For example, if there is 100m snow with some but small wetness, L-band can detect aquifer as penetration depth is higher, while higher frequencies cannot see the wet aquifer layer.

19. The results need to be compared with in-situ values or other algorithms. There are some other algorithms using L-band (refer to my comment number 10). Same authors, I believe have done a similar thing for Greenland.

---

## Referee Comment (RC2)

**Reviews for: "The sensitivity of satellite microwave observations to liquid water in the Antarctic snowpack" by Picard et al., 2022**

**1   Description**

In the article "The sensitivity of satellite microwave observations to liquid water in the Antarctic snowpack", the authors discuss and provide new insights into the theoretical basis of passive microwave liquid water detection algorithms over shelf ice.

First, the authors compare different permittivity models for wet snow and discuss their applicability for microwave emission modeling. Further, based on a Monte-Carlo approach, the authors retrieve the dry snowpack properties at 8 different sites on the Antarctic shelf ice and then simulate the sensitivity of different microwave channels to the appearance of liquid water in snow. The results are compared to satellite observations.

Several experiments are performed analyzing e.g., the sensitivity of microwaves to liquid water layers in snow at different depths and with different magnitudes.

In the last part of the manuscript, the authors give recommendations how their results could be used to improve existing or develop more advanced passive microwave melt detection algorithms and for an improved evaluation of firn models.

In general, this article provides interesting and new results which are of high scientific relevance and can help to improve existing or develop more advanced passive microwave melt detection algorithms. The article is clearly structured and the individual sections and experiments are well motivated. However, some of the figures need to be improved. A few statements are not clear or hard to reproduce from the figures. Additional clarification and information is sometimes required. Therefore, I recommend the article to be published after minor revisions.

**2   General Comments**

I have two general comments. 1) the manuscript would benefit from improving some of the figures. Some of the lines in the figures are hardly visible or distinguishable from other lines. I recommend that the authors check all figures again and increase the line width when appropriate or choose different colors for different lines. Further details are provided in the specific comments.

2) Overall, the modeling setup is well described with the right amount of detail. This method is a very nice approach highlighting the potential strength of snow emission models like smrt. I think it could be very useful for other scientists who are interested in using the model or the method if the authors would provide a more detailed description of the experiment setup in the supplement (e.g., adding the initial range of all input parameters, differences of the different sites, similarity of the $\approx 100$ output profiles,...)

**3 specific comments**

L10: *and on the site* → Here I would specify what controls the layer depth (e.g., density, grain-shape, terrain properties)

L18: *climatic indicator* → climate indicator

L47 *when it reaches* → It reads like you mean a threshold for the surface but instead it is a threshold for the brightness temperature

L55-56: What about snow redistribution? I believe this could be more important than snowfall events and also much harder to predict/simulate

L92: *...presence of aquifer Montgomery...* → presence of aquifer, Montgomery

L95: it would be helpful if the authors would provide a map with the different study sites marked

L100: *The Soil Moisture...* → Observations from the Soil Moisture...

L106: Do you average the observations from the incident angle range?

Table 1 Caption: *AWS names are from (Jakobs et al., 2020)* → AWS names are from Jakobs et al. (2020)

Table 1 Caption: You need to specify which temperature is used (i.e., 2-m or skin-temperature?)

Table 1 Caption: *melt days are from AMSR2 19 GHz H-pol channel.* → I assume a published algorithm is used? (Then the reference is missing) Or is it based on the 20K threshold? (Then why did you chose this channel?)

L128-130: Earlier, you wrote that you use SMOS data from 50 to 55°. I was wondering, do you then also simulate the SMRT output for this incident angle range?

L134: *microwave* → microwaves

L175ff: Based on Figure 1, I cannot reproduce the authors argumentation that the MEMLS V3 model behaves close to the coated spheres model in the high water regime. Aren't the differences between these two model much larger than e.g., to the Colbeck 1980 (Pendular) model? In addition, it is really hard to distinguish the different models in the low water regime, which would be the most important part in case one wants to know the minimum water content detectable by microwave observations. I suggest adding a zoomed-in version of Figure 1.
Also, would in not make sense to, in addition to the Hallikainen 1986, use a model which predicts higher changes in permittivity (e.g., the Colbeck 1980 (Pendular) model), to have an idea of the possible ranges of the sensitivity to liquid water?

L192: Sensitivity of what? The brightness temperatures? I would've assumed that (small-scale) surface roughness variations can have an impact at least at H-Pol. Maybe the authors could provide a rough number for the impact of surface roughness variations at the snowpack brightness temperatures

L201: I assume that the 2-m air temperatures is used?

L204ff: I wonder if the authors would have included L-Band data, how would the optimal snowpack change? I guess the other frequencies would then look worse?

L2010: *400−910* → wrong symbol. Also later in the document, − and - are sometimes mixed up. I suggest the authors to carefully check throughout the document

L218: $d/2–2d$ → This reads d/2 minus 2d but I guess you mean d/2 to 2d

L221: The citation should be earlier

L230: I was wondering, since you compare point simulations with large-scale satellite observations, have you considered the effect of slopes in the footprint? I guess there might as well be slopes on the Antarctic shelf-ice within one satellite pixel. If they have an impact on the observations, your retrieval method would compensate with changed snow properties and thus the snowpack might not be representative for the specific satellite pixel.

L250: *distinctively low SMB* (Table 1)

L249-250: According figure 3, the snowpack at Amery has the highest correlation length (i.e., largest grains) of all sites at 8 m (which is around the e-folding depth at 6 GHz, I guess). This could (partly) explain the relative low brightness temperatures observed at this site.

L262-264: Here, you discuss the dependence of H-Pol on the ice layer density. However, I miss the relation/implication of this for the brightness temperatures at the different site. Since you use the ice layer density as an variable in you model input, it would be nice to also discuss how (qualitatively) this variable is different at the different sites and how this relates to the H-pol observations.

L272: For clarification, is the RMSE calculated from the set of 100 snowpacks for each site?

L275-282: I was wondering, did you use the same scaling factor for all sites? From figure 2, it looks like you sometimes over- and sometimes underestimate the observations at L-band, so wouldn't a variable scaling factor make more sense?

L295-299: For Amery, larsen C and Larsen B, the correlation length at 8 m is much higher than at 20 m. I have difficulties to find a physical explanation for that. Could this is an artifact of the choice of the depth of the tie points. Since the e-folding depth at 6 GHz is well below 20 m depth (based on the examples you provided for Baudouin), the contribution of this tie point to the model result is very low. If that's the case, this should be shortly discussed in this paragraph.

Figure 3: The color for 0 cm is hard to see. I would recommend to increase opaqueness here

Figure 3, caption: *grain size* → correlation length

L310-311: here you write that the surface density is $220 Kg m^{-3}$. However, according to figure 3, this value varies between 200 and 400.

L314: *imaginary part of the water permittivity is extremely high (Fig. 1)* → I'm not able to see from figure 1 due to the large range of the permittivity.

L316-319: What does *pixel.day* mean? Which AMSR-2 observations are you using? Only for the Baudouin grid cell or for the whole Antarctic shelf region?

L320-324: I have difficulties reproducing the numbers given in this paragraph. Are you still describing the results shown in figure 4 at 19 GHz? E.g., none of the H-pol values in figure 4 (a-c) reach 260 K and $\Delta$Tb at 19 GHz, H-pol seems to be less than 60 K

Figure 5: This figure needs to be revised since different lines are very hard to distinguish. One option could be to show less frequencies and then show an "errorbar" plot with the $\sigma$ or $2\sigma$ spread as shaded contours.
Figure 5: There seems to be a step at several frequencies e.g., at 19 Ghz H-pol between $12.5\,Kgm^{-2}$ and $15\,Kgm^{-2}$ total liquid water. I was wondering what is the possible reason for that?

L345-350: Another conclusion would be use V-pol over H-pol since its much less affected by surface processes while having sensitivity to liquid water content

Figure 6: Same as figure 5, the contrast of the curve for the highest standard deviation is too low

L373-374: Mention that the figure only covers the 1st regime

L376-377: I don't see how the Zwally algorithm mitigates the problem of high (winter) brightness temperatures, since they are using a fixed threshold (30 K) which would not be reached for e.g., Halvfarryggen

Section 4.2.4: While it is interesting to discuss the shortcoming of the H86 formulation, it would be also of value to compare the results of the (reasonable) selection of models shown in figure 1 to get an idea of the spread of the solutions.

Figure 8: decrease the Y-extend

Figure 9: The lines for 1 and 2 cm are hardly visible. Why is in figure 9b only H-Pol shown?

L405-406: *We find that varying the thickness of the wet snow layer has little influence on the results if the total amount of liquid water is fixed *

L409: *for small amounts of water*

L411: Given the statement "many authors" I at least expect some references

Figure 10: It would be nice to add the dry snowpack brightness temperatures (e.g., as triangles at the right end of the different lines)

L403: *(decreasing in exponential shape)*

L434-435: *especially at 37 GHz, due to the accumulation of fresh, fine-grained, snow over the summer, metamorphosed, coarse-grained snow.* I have problems understanding the last part of this sentence.

Figure 11: (Caption) *Time-series of AMSR2 brightness temperature at Wilkins at V-pol.* The figure would benefit from adding the ERA5 2m air temperature so one can easily assess when melting starts/ends

L455: *with the depth of the wet snow layer*

L455: *and weaker than at the*

Figure 12: Add polarization shown here.

L456: *because the dry snowpack is already close to a black body *

L459: *features* → feature

L461-463: To me, it looks like at 6 and 10 GHz, the results at V-Pol are well comparable and only at the higher frequencies, they strongly differ

L475: *brightness temperatures*

L487: *characterized by  very low*

L497:500: This sentence is quite lengthy and written in a somewhat colloquial language. I would recommend to rephrase this paragraph and maybe split it into several sentences

L504: *triggered* → triggering

Figure 15: Use solid lines for V-Pol threshold

L510: *overall lower sensitivity of L-band compared*

L511-512: I'm not sure I understand what you are referring to here. From Figure 16, it does not look like the signal at V-pol becomes saturated at $30\,kgm^{-2}$, also I do not see a maximum at H-pol at 14 but rather at $8\,kgm^{-2}$. Please explain what exactly you are describing here

L521-522: Earlier you stated that a threshold of $10\,K$ brightness temperature differences would still be way above the noise level. Refining this statement (so what would be the minimum acceptable threshold) would help the interpretation of the modeling results

L527: *(213 K at H-pol)*

Figure 17: The different behavior of V-pol and H-pol needs to be addressed in the text. Why is there a maximum at H-pol when the wet layer is buried just below a snow layer?

L563-563: Since the snowpack evolution during the wet season is not (or only partly in experiment 3) covered by your simulation setup, this statement is not really a finding of this study but more a general problem.

L571: I don't think, using V-pol would "avoid" the problem of not detecting wet snow with high water content. It would rather mitigate the problem towards slightly higher water contents

L584: *particularly on the ice shelves*

---

## Author Comment (AC1)

The authors would like to thank the reviewer for their comments and feedback. Our responses are presented in blue.

Reviewer #1

Synopsis:

In this paper, the sensitivity of microwave brightness temperature to snow liquid water content at different frequencies from L-band to Ka-band has been investigated. Considering a recent interest in multi-frequency snow monitoring, authors have done a good job in drafting this paper. However, there are some issues that need to be addressed before this manuscript can be published in this journal. Please see below for my major comments.

Comments to the Author:

1. In line 66, where you mentioned that "... 4) when the surface becomes extremely wet (i.e. a saturated water layer, running water, surface ponding), the microwave signal is affected to the point that melt detection may become impossible despite the surface being obviously wet.", can you please explain why it becomes impossible? When it is very wet to the point to the point that a microwave signal is saturate (depending on the frequency of operation), a melt event for that specific season would be detected, but there may be a delay in detection (depending on frequency), or even uncertainty in the correct absolute value of liquid water content, as one frequency signal gets saturated, it cannot retrieve any further excessive wetness. And here is when multi-frequency detection becomes valuable as you mentioned. That said, please correct me if I'm thinking wrong.

At this stage of the paper (the introduction) we can not address in full detail this complex topic, the reason why the sentence is a bit vague. We will slightly reformulate the sentence to give a better but still brief explanation, considering that the topic is addressed in detail further in the paper, in Section 4.5 about Experiment 4 and in Figure 15. "When the surface becomes extremely wet (i.e. a saturated water layer, running water, surface ponding), the microwave brightness temperature decreases -- because open water surfaces have a low emissivity (Comiso and Cavalieri 2003) -- up to the point that melt detection may become impossible despite the surface being obviously wet."

Furthermore, we do not understand the comment well, because the sentence is here about the "water saturation" (water is so abundant that no air remains in the snow) or pure water which is different from "microwave saturation" addressed by the reviewer (i.e. the brightness temperature at V is stable from some small water amounts). The former situation occurs for very high water content and in such a case, the signal at H polarization is low at all the frequencies (as on the ocean), the penetration is extremely short at all the frequencies. We don't see the benefit of the multi-frequency direction in this case.

2. In line 80, where you mentioned "This is only possible when the snowpack is dry, before or after the melt season.", isn't it different to use before or after melt season as a good dry snow condition? A snowpack that went through a longer winter season, will become closer to a more ideal dry snowpack; hence, a snow condition more to the winter season (further away from last day of melt season) would

be a better dry snow condition. Can you please explain this a little bit here and where it's necessary in the paper?

We agree with the reviewer that the snowpack is different before and after the melt season. The 'ideal' snowpack for our purpose is probably the winter snowpack for the earliest stages of the melt season and the autumn snowpack for the late stages when the snowpack is ripped. In practice we mostly use the winter snowpack in this paper and do not consider the progressive changes over the course of the season, which could be a further refinement of our study.

However, it seems that the reviewer's comment is raised because 'dry snowpack' is sometimes used for a snowpack that has never been subject to melt. Here and throughout the paper, we use "dry" to strictly mean "actual absence of liquid water" and we use the more subjective terms "winter / autumn snowpack" in the following to distinguish the typical snowpacks before and after the melt season. We propose to change "dry snowpack" into "dry snow" here but we keep "dry snowpack" in the following.

3. In line 153, where you mentioned "A third regime described by Colbeck ...", can you please add this sentence to the end of paragraph? It's too small to be a paragraph on its own.

It is indeed small, but it does not fit well in the previous paragraph which is about the funicular regime. We propose to completely remove this non-essential statement.

4. In line 234, you mentioned "Water was always added by filling the air pores, which means that the ice mass (i.e. the dry snow density) is kept constant.", can you please why it has been decided to keep the density constant? Shouldn't the snow density change with snow melting, at least for the tope layers, where usually undergo metamorphism?

It is correct that the total snow density is likely to change during melt because of the physical processes of metamorphism and densification. Other properties of the snowpack are also likely to be affected, adding more variations on Tb on top of what we have simulated in this paper. We have not taken these physical processes into account because 1) we want to investigate the variations of Tb as a function of a single effect, the addition of liquid water only, all other properties (density, grain size, …) being constant for the clarity of the analysis, and 2) these processes are complex and would require running an advanced snow models (e.g. Crocus or SNOWPACK), adding complexity to this study. Here all the presented results are only due to the liquid water, not to other concomitant structural and microstructural changes in the snowpack.

5. In Figure 2, please , mention in the caption the period (months of year) observed brightness temperature.

This will be added.

6. In line 254-255, where you mentioned "For instance, at H-pol, the brightness temperature is low and close to that at 6 GHz on Amery and Larsen C, whereas it is much 255 higher and close to that of 37GHz in the other sites. The reason for this is not clear.", first of all, it is very important when these measurements have been performed. For example, some years, some ice shelves can undergo more metamorphism, which introduces more distinct vertical layer boundaries, which in turn decreased the

brightness temperature drastically, mainly in H-pol. Also, surface scattering can decrease TB if it's comparable to the wavelength, and maybe that's why L-band is close to 6GHz in Amery and Larsen C. In general, as frequency increases, the applied electric field changes polarization faster and forces the dipoles in the dielectric to change directions faster as well. Hence, dielectric loss increases. In addition, as the frequency increases, E-field changes polarization faster than the dielectric dipoles' relaxation time, which in turn decreases the displacement electric field in the material and decreases the dielectric constant. Hence, TB increases (less reflection). Perhaps, by comparing different measurement during different periods of year (or even same months but different years), you may be able to demystify why L-band behaves like this.

Thanks for the comment, what is suggested could be very relevant but if one wants to accurately estimate the temporal Tb trends in a specific site where accurate information (or hypothesis) of snow structure is available. Nevertheless, this is out of the scope of the paper which intends to contribute to a better understanding of melt detection using an approach that balanced precision and simplicity. In our case the precision is not required because our simulations with wet snow do not depend too much on the dry snow properties. The Tb winter inter-annual dynamics, as evoked by the reviewer, is a different and more elaborated topic and it would certainly require a different paper, with different goals and approach. This is the reason why the section on the dry snowpack is no more than 1 page long and that we do not address the dynamics of the snowpack.

7. In line 256, it was claimed that V-pol suggests that scattering increases with number of melt days. Can you please explain why? This is contradictory with the statement in line 262, where you mentioned that H-pol and V-pol are controlled similarly by scattering. Your claim in line 256 could be right, but it definitely needs more proof and validation.

We propose to extend the paragraph in the following way:

"In general, V-pol brightness temperature at Brewster angle is mainly driven by volume scattering and snow temperature. The observed decreasing trend with the number of melting days (e.g. a 40\,K decrease at 37\,\unit{GHz}) suggest that scattering is increasing with the number of melting days" and similarly change the paragraph on the H-pol, as addressed in the next comment (8).

8. In line 266, where you mentioned H/V ratio becomes larger mainly due to the layering, can you explain this? Assuming that there is no attenuation, and no surface scattering, are we still seeing this effect? Usually, H and V observations becomes close to each other when there is enough surface scattering to make the signal to loose its coherency, then it becomes polarization- independent.

We will reformulate this paragraph in the same way as the previous one about V-pol (see also Reviewer #2 comment).

"In general, the H-pol brightness temperature is more complex because it is in part controlled by snow scattering and snow temperature (exactly as V-pol) and in addition, it is sensitive to the surface density and the vertical density fluctuations in the snowpack (layering). The ice layers decrease the brightness temperature at H-pol due to the reflections on the high dielectric contrast between snow and ice in the upper part of the firn \citep{montpetit_2013}. The variations in V-pol and H-pol are correlated and of similar amplitude only if the ice layer effect is negligible. "

The physical reason for this behavior of H/V with the ice layers is well documented (E.g. Montpetit et al. 2013).

9. Please also explain why in figure 2, V-pol decreases and then increases, while H-pol monastically decreases by increasing the number of melt days.

The two reformulated paragraphs in this section interpret the variations observed in Figure 2. However, we don't see the simple variations depicted by the reviewer, to us, the variations are more complex. Our explanations in this section are therefore limited to a few points that seem to us robust enough, but we do not understand all the variations in the Tb and in the retrieved parameters.

10. Can you please compare your retrieved snow density with that of other available methods, such as "M. Mousavi, A. Colliander, J. Z. Miller and J. S. Kimball, "A Novel Approach to Map the Intensity of Surface Melting on the Antarctica Ice Sheet Using SMAP L-Band Microwave Radiometry," in IEEE Journal of Selected Topics in Applied Earth Observations and Remote Sensing, vol. 15, pp. 1724-1743, 2022."?

As stated in our manuscript, "While it is tempting to analyze the retrieved properties of those snowpacks, it is important to recall that the problem is under-determined and the snowpack representation simplified. Many equifinal sets of parameters give similar brightness temperatures, despite they may depict quite different snowpacks from one another, and from the real snowpack as well \citep{beven_1992}."

It is not our ambition to retrieve snowpack properties that would be comparable with other methods or in-situ data. Here, our objective is to quantify the melt effect by using a simplified but realistic representation of dry snow pack but the aim is not the retrieval of snow density.

In fact, we don't believe it is possible to retrieve density (and the other properties) with a useful accuracy for geoscience using microwave data only. As explained in our paper, the retrieval problem is under-determined when considering four layers and three unknown properties per layer. To obtain a well-constrained problem, the snowpack would need to be oversimplified with 1 or 2 unknowns, that is using a single layer, or multiple layers but assuming some constant values from some properties and neglecting the ice layer. The consequence would be 1) easier retrieval of a density value, but 2) this estimate would be uncertain, carrying all the neglected effects (notably layering) and would be method/assumption-dependent. This is certainly the case for the study cited by the reviewer.

To avoid the oversimplification of the snowpack in our setup, we decided to keep the problem under-determined, but as a consequence, we don't obtain a single value for the density. Instead we obtain a probability of possible densities and in general unfortunately, the range of possible densities is very large... too large to be useful in a comparison or for geophysical applications.

This is why we think that our "realistic" snowpack is only realistic from a microwave point of view, not from a geophysical point of view.

11. In line 318, what do you mean by "pixel.day"? Can you please explain this unit to avoid confusion?

This unit is frequent in studies on melt and passive microwave (cumulative melt area) but we will reformulate to avoid it:

"In reality, such high brightness temperatures are rare, they can be found only 347 times in the full daily AMSR2 19\,\unit{GHz} records gridded at 12.5\,\unit{km} over 9 summers (2012--2021). For comparison the number of times melt is detected on this same grid and for this same period is $2.5 \times 10^6$."

12. Please specify incident angle in Figure 4, and wherever it's needed.

The value is now indicated in the method section and is the same for the whole paper.

13. Can you please compare your result for snow wetness with other algorithms? You can convert it to liquid water column using density and thickness of wet snow layer.

It is not clear to which part of the manuscript this is referring to. The aim of the paper is to better understand melt detection at the different frequencies, we do not intend to retrieve snow wetness.

14. In Figure 5, please specify in the caption that solid curve is V-pol and dashed one is for H-pol.

Done.

15. In line 404-407, it was claimed if the total liquid water content is fixed, wet snow layer thickness has little effect. First of all, can you please write an expression to relate volumetric liquid water content to the total liquid water content? If the thickness is normalized in the total liquid water content, the thickness should have zero effect with fixed total liquid water content, in other words, as thickness increases one would expect the signal to lose more energy with exponential term depend on the thickness and the imaginary part of propagation constant. However, if the imaginary part is normalized by depth, and kept constant, the exponential term would be constant. That said, I believe a bit more explanation with some equations would avoid any confusion.

We will reformulate the paragraph and add the equations, as follows:

"The previous results were obtained with a fixed wet snow layer of 10 cm. Figure \ref{fig_aws19_experiment2} reports a few results for different thicknesses of wet snow $h_w$ (for Roi Baudouin and at 6 and 19\,\unit{GHz} only) considering two predictor variables (i.e. the variable on the x-axis): the total amount of liquid water $TLW$ or the volumetric fraction of water $\theta_i$. These variables are related by: $TLW = \theta_i \rho_\textrm{water} h_w$ where $\rho_\textrm{water}$ is the water density."

16. In line 418-420, where it was claimed that total water content has more effect than volumetric water content, it is very confusing. It looks like there are two different parameters. This needs to be clearly explained.

We acknowledge that "main/second driver" was misleading, our purpose was to explain why the total amount of liquid water is a better predictor than the volumetric water content if the wet layer thickness is not known  or is varying. We reformulate the conclusion to fit closely to the paper's aim.

"In conclusion, plotting brightness temperature variations as a function of the total water content (\unit{kg\,m^{-2}}) allows a better generalization of our results for other thicknesses than if the volumetric water content was used. For this reason, we generally present our results in total water content."

17. In line 470, where you mentioned "... saturated layer exceeds a quarter of the wavelength ...", why quarter wavelength? It should depend on the penetration depth. Can you please explain where does this quarter wavelength come from?

This is the (approximate) value from which the "coherent effect" disappears and when the reflection of the layer can be calculated with the (incoherent) Fresnel's coefficients. Under this limit the reflection is not as strong. We will add a reference to Wiesmann and al. 1998 which is specific to snow, but this effect is very common in electromagnetism and has many names (see e.g. https://en.wikipedia.org/wiki/Thin-film_interference).

18. In Lines 519 – 521, where you mentioned "This is in apparent contradiction with the results in Fig. 17 showing the brightness temperature at L band with a 30 cm thick wet layer at increasing depths (Experiment 3)." If there is a very wet layer, such that the thickness is greater than the penetration depth of the frequency of operation, it becomes a very reflective layer, and drops the TB, no matter what depth this layer has been resided. S, it is very confusing that you are claiming it is in apparent contradiction with (Mätzler, 1996; Macelloni et al., 2016, 2019) for L-band. For example, if there is 100m snow with some but small wetness, L-band can detect aquifer as penetration depth is higher, while higher frequencies cannot see the wet aquifer layer.

Our simulations show that for wet snow with a moderate amount of water, this layer behaves as a black body, not as a high reflector. We will change "wet layer" to "wet snow layer" everywhere it was confusing and we will add the total amount of water (6.5 kg/m2 corresponding to the maximum of Tb H-pol in Fig 16) which was not explicit.

The reviewer describes the situation of a saturated layer of water, and indeed the theory suggests that such layer behaves as a reflector (lowering Tb) and would be seen at much greater depth (~100m at L band). This is the case of the sea water under ice-shelves for instance, visible for thin ice-shelves or buried lakes. We have not included this case in the paper because we have not found real observations for this situation caused by melting.

19. The results need to be compared with in-situ values or other algorithms. There are some other algorithms using L-band (refer to my comment number 10). Same authors, I believe have done a similar thing for Greenland.

It is not clear which part of the manuscript this comment refers to. Unless we misunderstand what is meant by "algorithm". Our paper is not about retrieval algorithms, the topic is a sensitivity analysis of brightness temperature to liquid water. Its intent is to be useful to build better algorithms in the future, but we do not propose any precise retrieval algorithm here.

---

## Author Comment (AC2)

The authors would like to thank the reviewer for their comments and feedback. Our responses are presented in blue.

**Reviewer #2**

1. Description

In the article "The sensitivity of satellite microwave observations to liquid water in the Antarctic snowpack", the authors discuss and provide new insights into the theoretical basis of passive microwave liquid water detection algorithms over shelf ice.

First, the authors compare different permittivity models for wet snow and discuss their applicability for microwave emission modeling. Further, based on a Monte-Carlo approach, the authors retrieve the dry snowpack properties at 8 different sites on the Antarctic shelf ice and then simulate the sensitivity of different microwave channels to the appearance of liquid water in snow. The results are compared to satellite observations.

Several experiments are performed analyzing e.g., the sensitivity of microwaves to liquid water layers in snow at different depths and with different magnitudes.

In the last part of the manuscript, the authors give recommendations how their results could be used to improve existing or develop more advanced passive microwave melt detection algorithms and for an improved evaluation of firn models.

In general, this article provides interesting and new results which are of high scientific relevance and can help to improve existing or develop more advanced passive microwave melt detection algorithms. The article is clearly structured and the individual sections and experiments are well motivated. However, some of the figures need to be improved. A few statements are not clear or hard to reproduce from the figures. Additional clarification and information is sometimes required. Therefore, I recommend the article to be published after minor revisions.

2. General Comments

I have two general comments. 1) the manuscript would benefit from improving some of the figures. Some of the lines in the figures are hardly visible or distinguishable from other lines. I recommend that the authors check all figures again and increase the line width when appropriate or choose different colors for different lines. Further details are provided in the specific comments.

Regarding Figures 5 and 7, we acknowledge that some curves overlap but this serves our intent to communicate the range or the variations rather than each individual curve. Figure 5 is designed to show the ensemble and the range spanned by this ensemble, not the members. Figure 7 shows the variations for many sites. Our goal here is precisely to highlight these inter-site variations to support the text in the Section describing this figure. The reader interested by a specific site can still zoom in (in pdf) but this is not the purpose of this figure.

Regarding Fig 6, Fig 9 and Fig 15, the readability is affected by a limited number of colors. In fact, it is due to our adoption of a common color scheme for plotting brightness temperature throughout all our recent papers since ~2020. Each AMSR channel has a specific color. This color scheme is color-blind optimized with large contrast. However, as a consequence, figures showing one or two frequencies only have low contrast. Despite this disadvantage we think that the figures are still visible.

Regarding Fig 1, it will be improved as suggested.

2) Overall, the modeling setup is well described with the right amount of detail. This method is a very nice approach highlighting the potential strength of snow emission models like smrt. I think it could be very useful for other scientists who are interested in using the model or the method if the authors would provide a more detailed description of the experiment setup in the supplement (e.g., adding the initial range of all input parameters, differences of the different sites, similarity of the ≈100 output profiles,...)

We will add the prior ranges of all input parameters in the text, and all the results for the different sites are in Figure 3 (the mean of the properties). Only the variability of the properties is not shown in the paper. It has not been studied in detail in fact but it is overall large (see Figure below of the probability distribution of all the parameters at a selected location). This is by design of our method (e.g. using less observations than unknowns) that aims at reproducing the microwave signature during winter time, not at retrieving the properties accurately. As a consequence we don't think we can learn a lot about the real snowpacks from the diversity of the profiles in the context of the present paper. This diversity is mainly a consequence of our choices rather than something real, different choices in the number of tie-points and type of properties would have led to different diversity. Using this method for another goal is certainly possible and interesting – e.g. to retrieve the snow properties to be used in comparison with in-situ or firn model outputs --, but it implies to reconsider these choices – e.g. take a simpler representation of the profile to let less free parameters and try to reduce parameters with correlated or anti-correlated impact on the microwave signal, in order to reduce the equifinality and increase the interpretability of the retrieved properties.

[Figure]

Distribution of the retrieval grain size (left column), density (central column) and ice layer number density (right column) for the four snow depths at Shackleton.

3. specific comments

L10: and on the site → Here I would specify what controls the layer depth (e.g., density, grain-shape, terrain properties)

We will reformulate as suggested:

"ii) the detection of a buried wet layer is possible up to a maximum 1 to 6 m depth depending on the frequency (6–37 GHz) and on the snow properties (grain size, density) at each site"

L18: climatic indicator → climate indicator

will be done

L47 when it reaches → It reads like you mean a threshold for the surface but instead it is a threshold for the brightness temperature

will be done

L55-56: What about snow redistribution? I believe this could be more important than snowfall events and also much harder to predict/simulate

We will reformulate the examples, adding "blowing snow".

L92: ...presence of aquifer Montgomery... → presence of aquifer, Montgomery

will be done

L95: it would be helpful if the authors would provide a map with the different study sites marked

We will insert the suggested figure  (Figure 1 in the paper)..

[Figure]

L100: The Soil Moisture... → Observations from the Soil Moisture…

will be done

L106: Do you average the observations from the incident angle range?

The data are already binned in this range.

Table 1 Caption: AWS names are from (Jakobs et al., 2020) → AWS names are from Jakobs et al. (2020)

will be done

Table 1 Caption: You need to specify which temperature is used (i.e., 2-m or skin-temperature?)

will be done

Table 1 Caption: melt days are from AMSR2 19 GHz H-pol channel. → I assume a published algorithm is used? (Then the reference is missing) Or is it based on the 20K threshold? (Then why did you chose this channel?)

We have used a published algorithm here and will add the reference for this algorithm (Torinesi et al. 2003). Torinesi et al. is in fact often close to the 20K to our experience.

L128-130: Earlier, you wrote that you use SMOS data from 50 to 55 ∘. I was wondering, do you then also simulate the SMRT output for this incident angle range?

We will change the text to indicate that 55° is used for both AMSR2 and SMOS in the simulations.

L134: microwave → microwaves

L175ff: Based on Figure 1, I cannot reproduce the authors argumentation that the MEMLS V3 model behaves close to the coated spheres model in the high water regime. Aren't the differences between these two model much larger than e.g., to the Colbeck 1980 (Pendular) model?

We implicitly meant relative to Hallikainen, 1986, but it was not clear. We will change this paragraph.

"In this study, we selected the MEMLS v3 formulation for the reference simulations because it is based on actual measurements, and has an intermediate behavior"

In addition, it is really hard to distinguish the different models in the low water regime, which would be the most important part in case one wants to know the minimum water content detectable by microwave observations. I suggest adding a zoomed-in version of Figure 1.

Here, we present the Fig. 1 with zoomed graphs. However, it seems not useful for the purpose of the paper w.r. to the extra space needed, because the differences between the formulations are not discussed in the paper (because we don't have clue on why these differences). The paper actually only uses MEMLS formulation.

The code to produce the figure will be made available and the interested user will be able to zoom in the graph very easily.

[Figure]

Also, would in not make sense to, in addition to the Hallikainen 1986, use a model which predicts higher changes in permittivity (e.g., the Colbeck 1980 (Pendular) model), to have an idea of the possible ranges of the sensitivity to liquid water?

We have added the figure here and the notebook (that will be made available upon acceptance of the paper) has the option to generate this graph. The difference is small compared to MEMLS v3 even though it features as expected a higher sensitivity, particularly at low frequencies (i.e. 6 GHz).

[Figure]

Same as Figure 4 but for the coated sphere permittivity formulation.

L192: Sensitivity of what? The brightness temperatures? I would've assumed that (small-scale) surface roughness variations can have an impact at least at H-Pol. Maybe the authors could provide a rough number for the impact of surface roughness variations at the snowpack brightness temperatures

We'd also expected some effect but only got a strong effect for the active simulations (not used in this paper). The use of IEM in SMRT in passive mode has not been explored in detail to our knowledge in the literature; we are not able to provide a reliable value. We propose either to remove the sentence and completely overlook this aspect, or keep it as "after preliminary tests performed with SMRT and the IEM rough surface model" which indicates that this result is weak.

L201: I assume that the 2-m air temperatures is used?

Yes, it will be added in the text.

L204ff: I wonder if the authors would have included L-Band data, how would the optimal snowpack change? I guess the other frequencies would then look worse?

Yes, it is explained in L285. Adding L-band in the optimization without adapting the snowpack representation indeed fails, and negatively impacts the higher frequencies. The reason is probably because L band is sensitive to different (big) objects that are unrelated to the snow grains. Jezek et al. 2018 mention the role of ice pipes in Greenland, and we are convinced that this also applies to Antarctica, despite a weaker melt in general.

Adding such big objects in SMRT is feasible in principle but would imply to add even more free parameters in our optimization method, increasing the problem under-determination. As a tradeoff, we have decided to focus on the higher frequencies but still perform some L band simulations, because of the high interest, with the associated adequate warnings in the text.

L201: 400−910 → wrong symbol. Also later in the document, − and - are sometimes mixed up. I suggest the authors to carefully check throughout the document

L218: d/2–2d → This reads d/2 minus 2d but I guess you mean d/2 to 2d

The symbol will be corrected (double - in latex) but we will reformulate to avoid the symbol "recommended number between d/2 and 2d"

L221: The citation should be earlier

We will reformulate " the effective sample size as defined in Martin et al. (2022) is estimated ~100"

L230: I was wondering, since you compare point simulations with large-scale satellite observations, have you considered the effect of slopes in the footprint? I guess there might as well be slopes on the Antarctic shelf-ice within one satellite pixel. If they have an impact on the observations, your retrieval method would compensate with changed snow properties and thus the snowpack might not be representative for the specific satellite pixel.

The slope on the ice shelf is usually very small (<0.5°) and the expected effect on the passive measurements is probably of the same order as a change in the incidence angle. We believe that this is a very small effect but are not aware of studies in the literature about this problem. It is true that the optimization would compensate for that, but not for a good reason.

We will add "The terrain is assumed flat" in the description of the method, next to the "surface roughness".

L250: distinctively low SMB (Table 1)

will be added

L249-250: According figure 3, the snowpack at Amery has the highest correlation length (i.e., largest grains) of all sites at 8 m (which is around the e-folding depth at 6 GHz, I guess). This could (partly) explain the relative low brightness temperatures observed at this site.

Yes, but it is more correct to say that our Bayesian method estimates a high correlation length because the observed brightness temperature is low.

Here in L249-250 we only mention the particular SMB at Amery and address the mechanistic link between the brightness temperature and the correlation length mentioned by the reviewer in L 255.

L262-264: Here, you discuss the dependence of H-Pol on the ice layer density. However, I miss the relation/implication of this for the brightness temperatures at the different site. Since you use the ice layer density as an variable in you model input, it would be nice to also discuss how (qualitatively) this variable is different at the different sites and how this relates to the H-pol observations.

We propose to reformulate the beginning of the paragraph, following the reformulation of the previous paragraph on V-pol to better explain how in principle the ice layers contributes to H-pol:

"In general, the H-pol brightness temperature is more complex because it is in part controlled by snow scattering and snow temperature (exactly as V-pol) and in addition, it is sensitive to the surface density and the vertical density fluctuations in the snowpack (layering). The ice layers decrease the brightness temperature at H-pol due to the reflections on the high dielectric contrast between snow and ice in the upper part of the firn \citep{montpetit_2013}. The variations in V-pol and H-pol are correlated and of similar amplitude only if the ice layer effect is negligible. Here we find that at the highest frequency (37\,\unit{GHz}), the H-pol variations are close to that at V-pol. The reason is that the microwave e-folding depth is about one meter (e.g. 1.3\,\unit{m} for Halvfarryggen, and 0.75\,\unit{m} for Roi Baudouin) and only a limited number of layers are crossed by the upwelling radiation over such a small depth."

The site to site differences in the retrieved ice layer number are addressed in the paragraph related to Figure 3 a bit further. These changes are certainly not sufficient to give a clear view of how H-pol and the number of ice layers vary from site to site, but this seems too difficult to achieve. In fact, we don't understand all these subtle variations.

L272: For clarification, is the RMSE calculated from the set of 100 snowpacks for each site?

Only with the best parameters. This is indicated in the sentence "The simulations with the optimal parameters".

L275-282: I was wondering, did you use the same scaling factor for all sites? From figure 2, it looks like you sometimes over- and sometimes underestimate the observations at L-band, so wouldn't a variable scaling factor make more sense?

We use a constant scaling factor. We will change the text to make this point clearer:

"(we found that an **overall** factor of ~2.8 is necessary)"

"The grain size is multiplied by 2.8, **for all the sites** and only for simulating the L-band brightness temperature"

Using a variable scaling factor would certainly improve the simulations in dry conditions, but we believe it would not make "more sense". We don't have a fully satisfying solution to improve the L band in a consistent way, but still believe the results in wet conditions are useful.

L295-299: For Amery, larsen C and Larsen B, the correlation length at 8 m is much higher than at 20m. I have difficulties to find a physical explanation for that. Could this is an artifact of the choice of the depth of the tie points. Since the e-folding depth at 6 GHz is well below 20 m depth (based on the examples you provided for Baudouin), the contribution of this tie point to the model result is very low. If that's the case, this should be shortly discussed in this paragraph.

The profiles of the properties are linear between the tie-points, so the 20 meter tie-point does have some effect up to the upper tie-point (8 m). But we agree that with an e-folding of 5.2m at Beaudoin Ice Shelf, the contribution of depths >10m is certainly weak, and as a consequence the Bayesian method returns a relatively random value which turns out to be low here but could be large (Bayesian experts would not call this 'artifact' because this is an expected behavior of Bayesian approach when no information is available to constrain the value).

We propose to amend the paragraph on retrieved parameters:

"The mean retrieved parameters for all the sites are shown in Fig. \ref{fig_parameters_experiment0}. Some general observations can be made despite the risk of compensation between parameters (equifinality). **It is also worth noting that the properties at 20\,\unit{m} are not always constrained by the observations, as for instance when the e-folding depth is only 5.2\,\unit{m} at Roi Baudouin. In such a case, the method returns a virtually random value for this depth.**"

Figure 3: The color for 0 cm is hard to see. I would recommend to increase opaqueness here

will be done

Figure 3, caption: grain size → correlation length

will be done

L310-311: here you write that the surface density is 220 Kgm −3 . However, according to figure 3, this value varies between 200 and 400.

This section is about Roi Baudouin, we will move this information before referencing the surface density (220 kg/m3).

L314: imaginary part of the water permittivity is extremely high (Fig. 1) → I'm not able to see from figure 1 due to the large range of the permittivity.

We will move the reference to fig 1 at the end of the sentence so it is now clear that the value is to be compared to 0.0017 which is extremely small compared to the range in Fig 1.

"In the first regime, the sudden apparition of water at the surface of the ice crystal sharply increases the snow absorption because the imaginary part of the water permittivity **is extremely high  compared to that of ice 0.0017** at 19 GHz (Mätzler 2006) (Fig. \ref{fig_permittivity})"

L316-319: What does pixel.day mean? Which AMSR-2 observations are you using? Only for the Baudouin grid cell or for the whole Antarctic shelf region?

We will reformulate this sentence "In reality, such high brightness temperatures are rare, they can be found only 347 times in the full daily AMSR2 19\,\unit{GHz} records gridded at 12.5\,\unit{km} over 9 summers (2012–2021) in Antarctica. For comparison the number of times melt is detected on this same grid and for this same period is \$2.5 \times 10^6\$."

L320-324: I have difficulties reproducing the numbers given in this paragraph. Are you still describing the results shown in figure 4 at 19 GHz? E.g., none of the H-pol values in figure 4 (a-c) reach 260 K and ΔTb at 19 GHz, H-pol seems to be less than 60 K

We have corrected to "240-250 K" and "60K"

Figure 5: This figure needs to be revised since different lines are very hard to distinguish. One option could be to show less frequencies and then show an "errorbar" plot with the σ or 2σ spread as shaded contours.

As mentioned above, the purpose of this graph is to show the overall ranges, not to distinguish each line. The message we want to transmit is that the general shape of the curve is the same as in Fig 4, that the variability at V-pol is much smaller than at H-pol in general, and it is even possible to see that the variability for each frequency is almost the same.

Presenting aggregated stats (mean and sigma) rather than the raw data (ensemble member) has also some disadvantages. This is mainly a matter of personal preference and interest, it seems difficult to satisfy every reader (and reviewer) here.

Figure 5: There seems to be a step at several frequencies e.g., at 19 Ghz H-pol between 12.5 Kgm −2 and 15 Kgm −2 total liquid water. I was wondering what is the possible reason for that?

This is due to the numerical method DORT. We will add a note to point to the problem addressed in the main SMRT paper:

"Note that the small slope changes (e.g. around 17.5\,\unit{kg\,m^{-2}} at 19\,\unit{GHz}) is a numerical artifact due to this increasing strong permittivity and how SMRT deals with the refraction in the DORT method \citep[details in][]{picard_2018}."

L345-350: Another conclusion would be use V-pol over H-pol since its much less affected by surface processes while having sensitivity to liquid water content

We don't understand how this differs from our last sentence L349.

Figure 6: Same as figure 5, the contrast of the curve for the highest standard deviation is too low

See our response 1 to the general comment of the reviewer.

L373-374: Mention that the figure only covers the 1st regime

The sentence will be changed: "Fig. \ref{fig_sites_experiment1} shows the brightness temperatures as a function of the total liquid water conten**t (for smaller amounts than in the previous figures)** for a homogeneous pixel for the best snowpack at every site at 19 and 6\,\unit{GHz}"

L376-377: I don't see how the Zwally algorithm mitigates the problem of high (winter) brightness temperatures, since they are using a fixed threshold (30 K) which would not be reached for e.g., Halvfarryggen

The Zwally algorithm is adaptive as it takes into account the inter-pixel variations of winter brightness temperature across the continent, which is a first step but as noted by the reviewer this does not solve all the problems. Torinesi algorithm developed a few years later is more adaptive, not only because it adapts the 30K but also (and more importantly we believe) it calculates the mean winter temperature every year.

Our sentence is to briefly illustrate the adaptability, it seems correct.

Section 4.2.4: While it is interesting to discuss the shortcoming of the H86 formulation, it would be also of value to compare the results of the (reasonable) selection of models shown in figure 1 to get an idea of the spread of the solutions.

We have shown the possible shortcomings of the H86 formulation because it seems important however, addressing the similarities of the other formulations becomes a detailed intercomparison experiment and seems overdetailed for the purpose of this section, especially because we have no objective argument to choose one or another. This would lead to a complex but inconclusive analysis.

Since we will provide the code to run the simulations and make the figures, it will be easy for the interested user to investigate this issue.

Figure 8: decrease the Y-extend

it is on purpose the same as in Figures 4, 5 and 6. We only made an exception for Fig 7.

Figure 9: The lines for 1 and 2 cm are hardly visible. Why is in figure 9b only H-Pol shown?

This figure is not perfect but the choice of color, intensity and line type is a compromise between the consistency of the color scheme throughout the paper and the specific purpose of this figure. The main purpose of this figure is to demonstrate that the total liquid water is a better variable than the liquid water content because in the second case, the results depend a lot on the layer thickness. The important point is that most curves overlap in Fig 9a but do not overlap in Fig 9b. The visibility is not great but this is also what helps to distinguish these curves from the others.

V-pol is not shown in Fig 9b because it would add even more clutter. We could remove V-pol from Fig 9a to avoid this dissymmetry, but in the end decided that this was not a serious problem.

We will add a remark in the caption: "V-pol is not on graph b for the sake of visibility."

L405-406: We find that varying the thickness of the wet snow layer has little influence on the results if the total amount of liquid water is fixed when varying the thickness

We will change to "We find that varying the thickness of the wet snow layer has little influence on the results if the total amount of liquid water is fixed" but disagree with adding "varying the thickness" in the same sentence as suggested.

L409: for small amounts of water

will be added

L411: Given the statement "many authors" I at least expect some references

will be added two examples: Tedesco et al. 2006 and  Naderpour, R., & Schwank, M. (2018)

Figure 10: It would be nice to add the dry snowpack brightness temperatures (e.g., as triangles at the right end of the different lines)

will be added, on the left for the consistency with the other figures.

L403: (decreasing in exponential shape)

will be corrected

L434-435: especially at 37 GHz, due to the accumulation of fresh, fine-grained, snow over the summer, metamorphosed, coarse-grained snow. I have problems understanding the last part of this sentence.

We will change to "due to the accumulation of fresh snow with fine grains over the summer layer made of metamorphosed snow with coarse grains."

Figure 11: (Caption) Time-series of AMSR2 brightness temperature at Wilkins at V-pol. The figure would benefit from adding the ERA5 2m air temperature so one can easily assess when melting starts/ends

We will adde ERA5 T2m and highlighted the temperatures above freezing.

L455: with the depth of the wet snow layer

we will add "layer"

L455: and weaker than at the

will add "weaker"

Figure 12: Add polarization shown here.

Will Add " The polarization is horizontal. "

L456: because the dry snowpack is already close to a black body even when dry

We will move "dry" to the beginning of the sentence as suggested.

L459: features → feature

We will reformulate "The brightness temperature at 6 and 10\,\unit{GHz} features ", to solve another problem, the number 6 at the beginning of the sentence was not well separated from the 13 of the figure number in the previous sentence.

L461-463: To me, it looks like at 6 and 10 GHz, the results at V-Pol are well comparable and only at the higher frequencies, they strongly differ

They are overestimated in all the cases, but we agree that this is quite subjective. We will remove this remark.

L475: brightness temperatures

will be done

L487: characterized by a very low

we will remove "a".

L497:500: This sentence is quite lengthy and written in a somewhat colloquial language. I would recommend to rephrase this paragraph and maybe split it into several sentences

We will reformulate the paragraph:

"Searching for a significant decrease in brightness temperature in the AMSR2 dataset have been unsuccessful even on shelves subject to ponding, as for instance on the north George VI Ice Shelf where a maximum of 15\% lake coverage was observed \citep{banwell_2021}, or in on the eastern Roi Baudouin Ice Shelf."

L504: triggered → triggering

it seems that triggered is correct.

Figure 15: Use solid lines for V-Pol threshold

solid lines are already used for the brightness temperature variations. We need a different line style, at least to make it clear what these horizontal lines are in the caption.

L510: overall lower sensitivity of L-band compared

"at L-band" will be added

L511-512: I'm not sure I understand what you are referring to here. From Figure 16, it does not look like the signal at V-pol becomes saturated at 30 kgm −2 , also I do not see a maximum at H-pol at 14 but rather at 8 kgm −2 . Please explain what exactly you are describing here

For the H-pol, there was an error in the text, the value is 6.5 kg m$^{-2}$. We have rerun the simulations for Figure 17 for 6.5 kg m$^{-2}$ instead of 14 kg m$^{-2}$.

For the V-pol as well the limit is smaller than indicated in the text and it is better to refer to a maximum rather than a saturation, because the signal slightly decreases for larger liquid water contents. This will be corrected.

L521-522: Earlier you stated that a threshold of 10 K brightness temperature differences would still be way above the noise level. Refining this statement (so what would be the minimum acceptable threshold) would help the interpretation of the modeling results

We will add the values to make the small difference  more explicit.  "This value is relatively close to the dry brightness temperature (211 K vs 207 K),".

At such a low level of differences, we can not precisely elaborate on the acceptable threshold. 10 K is certainly acceptable to our opinion, but 4 K, we don't know. It depends on where and on the precise way the detection algorithm is working.

L527: (213 K at H-pol)

We will add H-pol earlier, after brightness temperature.

Figure 17: The different behavior of V-pol and H-pol needs to be addressed in the text. Why is there a maximum at H-pol when the wet layer is buried just below a snow layer?

This is addressed from line 526, but it was not clear because "H-pol" was not mentioned (see previous comment). This will be made clearer.

L563-563: Since the snowpack evolution during the wet season is not (or only partly in experiment 3) covered by your simulation setup, this statement is not really a finding of this study but more a general problem.

Experiment 3 shows that the maximum detection depth is slightly lower in autumn than in winter in most cases (note that the caption of Fig 3 was wrong, we have inverted dark and light). We acknowledge that the results are not marked (not as marked as we would expect), but this statement is based on this finding.

L571: I don't think, using V-pol would "avoid" the problem of not detecting wet snow with high water content. It would rather mitigate the problem towards slightly higher water contents

We will change avoid → mitigate.

L584: particularly on the ice shelves

will be corrected.

---

## Author Comment (AC3)

The authors would like to thank the reviewer for her comments and feedback. Our responses are presented in blue.

**Reviewer #3, Angelika Humbert**

The manuscript is concerned with a study of passive microwave sensing of the water content of snow/firn in the Antarctic. It contains high level extensive simulations of the brightness temperature for a variety of frequencies that correspond to satellite missions. In short – it is a fantastic work. The main critics are to explain the microstructure that is used for the simulations, to present the results in a way that not only radiative transfer modelers understand them, but also firn microstructure and hydrology modelers.

- In general it would be very good to show both, total liquid water content and volumetric water content in all graphs. This can certainly be achieved easily and would be super helpful for people who are working with modelling of snow/firn hydrology that think more in terms of volumetric water content.

We acknowledge that using a common variable used by the community (firn, snow and microwave) would be better, but the entire point of Section 4.3 (experiment 2) is to show that the simulations strongly depend on the layer thickness if the volumetric water content were used as the predictor, i.e. as the x-axis in Figures 4, 6, 7, and others. This is not our choice but a consequence of the physics of the radiative transfer, the brightness temperature is mainly driven by the total absorption in the case of a wet snowpack, and so on the total liquid water.

In principle we could add the volumetric water content as the x-axis or as a secondary x-axis (on the top x-axis) in those figures and write the layer thickness in the caption (usually 10cm). However, there would then be a risk that some readers will take these volumetric water content values without the layer thickness, and apply our results in a different context with a different layer thickness. That would be incorrect. For instance saying "brightness temperature saturates from xx % volumetric liquid water content" is meaningless unless the layer thickness is given. Several papers in the literature interpret volumetric liquid water content (often also called snow wetness) without also stating the layer thickness (the value is hidden somewhere in the methods section) and we believe that this is not a good practice anymore. This is the reason why, despite the wide use of volumetric water content in our community, we use the total liquid water here.

The ad hoc conversion from total liquid water to volumetric liquid water is easy, and we will add the equation explicitly to make it easy to translate the total liquid water to a pair: volumetric water content + layer thickness.

- I am missing a section that shows how well the parameter optimization is doing for the particular frequencies. This should be included in the next version of the manuscript either as a subsection of an appendix. But it is important to demonstrate the performance of this.

If we understand the comment correctly, this question is addressed in our Section 4.1 and Figure 2 (Fig 3 in the new future version).

- The manuscript needs to improve on building a link to microstructure observation and modelling. As an example 'we selected the exponential microstructure representation' is not enough for understanding what type of microstructure is chosen.

We will add the reference to a new paper (Picard et al. 2022), that was not published when this paper was submitted. This new paper demonstrates that the choice of the microstructure is a second order problem in the microwave domain, as opposed to what was believed before. For this reason using exponential or another microstructure is not critical for our simulations, as long as the same "microwave grain size" (as defined in Picard et al. 2022) is used with all the microstructures. Here we use the term correlation length with the exponential microwave which appears to be exactly the same as the "microwave grain size".

We will add "Very similar results would be obtained with a different microstructure representation as long as the same microwave grain size is taken for all microstructures as demonstrated in Picard et al. 2022".

- The way the temperature is simulated is problematic. I think the results could be substantially improved by using temperature simulations from snow/firn models. These simulations are provided by regional climate models such as RACMO, MAR and others. Given that the temperature is such a crucial parameter in the modelling, more efforts to get the simulated temperature of the snow/firn right are important.

It is true that the brightness temperature directly and linearly depends on the physical temperature profile and any uncertainty on the physical temperature has an impact on the properties estimation. However, we disagree with the fact that our temperature profile is problematic, given our objective. Here we only need a broad information on temperature (i.e. the averaged multi-year winter temperature using) to obtain "realistic" properties of a simplified multi-year averaged snowpack.
Even if the estimated properties were imprecise (and they are at least because the microwave observations are insufficient to constrain the snowpack even with perfect temperatures), our sensitivity analysis remains realistic and usable. By devising this estimation method, our goal was to get a bit closer to the reality than in previous studies where the snowpack was purely synthetic and oversimplified, but not to reach an accurate description or to retrieve useful properties for other goals than microwave simulations.

We believe that using regional climate models, which have not been extensively validated for the firn temperature to our knowledge, is too complex, relatively to the benefit we could get, given our final results (i.e recommendations for advanced melt detection algorithm). Our simple approach has the advantage of being readable and reproducible.

- The effect of heterogenous pixels: the coarse resolution of passive microwave sensors/missions, make it likely that the brightness temperature of a particular pixel is a mixture of different snow/firn/ice properties. This could be overcome by incorporating high resolution radar imagery, such as provided by Sentinel-1. In 10-30m resolution the homogeneity of a passive microwave pixel can be assessed. For the purpose of this study, wither pixels could be excluded that are not homogeneous or they could be characterized. To this end not a major effort is necessary, as no time series is required (in a first step) but it could help substantially to understand the differences between simulated and observed brightness temperatures.,

We agree with the reviewer that the heterogeneity of the melt and of the snow in 25 km wide pixels is an interesting topic and it could be addressed with Sentinel 1 or ASCAT. However, our goal here is to obtain "realistic" properties for an "averaged" snowpack over the 25 km resolution of the passive microwave observations (and over multiple years as explained in the previous comment). The fact that the observed brightness temperatures are the average emission of many different snowpacks within a pixel is in fact an advantage for this goal, not a disadvantage, because we want to estimate a representative "average" snowpack for the area rather than a particular snowpack. Our results are representative of the scale of the passive microwave observations.

- The manuscript lacks plots or diagrams showing the SMB, in particular melt rate in the study sites. This would help substantially to understand the results.

We don't understand the problem that would be solved with SMB plots or diagrams. The SMB values are given in Table 1 for each site, and the melt rate at most selected sites is available in the paper Jakobs et al., 2020, which we refer to on multiple occasions in our paper.

Line 35: either peninsula or Antarctic Peninsula

will be done

Line 107: +-2K.

We don't understand the comment.

Line 164: 'They are likely invalid for high contents' Contents of what? Elaborate more on what basis is that stated.

"water" will be added.

We don't have much information to elaborate on. The sentence will be reformulated to a more neutral statement: "Their validity for high water contents is unknown."

Line 192: property profiles

We will change to "the profiles of the properties"

Line 204: what are unknown tie points?

We will reformulate the sentence: "for 12 unknowns (3 properties at 4 tie-points)".

Line 235: The why is the seasonal temperature set to 273K? This does not seem to be appropriate.

Our sentence was misleading, we don't pretend the season temperature is close to 273K, but we want to generate a temperature profile that is realistic for the summer period, and we do so by setting Ts=273K which controls the near surface temperature.

We will reformulate by avoiding the term seasonal temperature:

"In addition to generate a temperature profile representative of the summer season, we apply the same method as in winter except that T_s is set to 273 K"

Line 254: 'Intermediate frequencies have intermediate behaviour' needs to be rephrased with clearer a statement.

We will change to "frequencies in between have an intermediate behavior"

---

## Referee Report (RR1)

**2nd revision for: "The sensitivity of satellite microwave observations to liquid water in the Antarctic snowpack" by Picard et al., 2022**

**1 General**

I thank the authors for carefully considering my comments and the extensive replies. Both of my general remarks are adequately addressed, despite I think figure 6 would benefit from showing the mean ± standard deviation for each frequency instead of 1000 individual, but this is up to the authors preferences. I have only a few minor comments left (see below).

**2 Specific comments**

**L87**: (Comiso et al., 2003)) → remove extra )

**L304**: when the e-folding depth is only 5.2mat Roi Baudouin. → frequency missing

**L312**: this increase in density... → I guess you mean "increase in ice layer density"?

**Figure 5**: The authors showed the same results using the coated sphere permittivity model in the response letter to the reviews. I think this comparison should be (shortly and qualitatively) discussed in the main manuscript, since the reader gets an idea of the impact of the model choice.

**Figure 11, caption**: add "The values of the dry brightness temperatures are marked by the triangles."

**L517**: "or in on the eastern Roi" → remove "in"

---

## Author Response (AR2)

We thank the reviewers for their supporting comments.

General
I thank the authors for carefully considering my comments and the extensive replies. Both of my general remarks are adequately addressed, despite I think figure 6 would benefit from showing the mean ± standard deviation for each frequency instead of 1000 individual, but this is up to the authors preferences. I have only a few minor comments left (see below).

We have changed Fig 6 according to the reviewer's suggestion, it is indeed clearer.

2 Specific comments
L87: (Comiso et al., 2003)) → remove extra )

Done

L304: when the e-folding depth is only 5.2mat Roi Baudouin. → frequency missing

We have added: "at 6 GHz, the lowest frequency used in the optimization"

L312: this increase in density... → I guess you mean "increase in ice layer density"?

Yes, it was misleading to use only 'density'. It is now corrected.

Figure 5: The authors showed the same results using the coated sphere permittivity model in the response letter to the reviews. I think this comparison should be (shortly and qualitatively) discussed in the main manuscript, since the reader gets an idea of the impact of the model choice.

We have added a sentence:
"The transition between the first, "absorption" regime and the second, "reflective" regime takes place around 0.75 and 1.75\,\unit{kg\,m^{-2}} at 37 and 6\,\unit{GHz} respectively. \cite{shi_1995} reported a slightly higher value of 3\,\unit{kg\,m^{-2}} for radar at C band (5.6\,\unit{GHz}). **Using a different formulation of the wet snow permittivity would change these values. For instance the coated sphere model (results not shown) has a higher imaginary part of the permittivity, which leads to even more rapid increases in the first regime, and a reduced decreasing rate in the second regime, because the absorption dominates even more than the scattering and reflection mechanisms."**

Figure 11, caption: add "The values of the dry brightness temperatures are marked by the triangles."

This is added.

L517: "or in on the eastern Roi" → remove "in"

Corrected.

Referee #3: Angelika Humbert

Many thanks for the efforts the authors took for improving the manuscript. I can fully understand that taking a new way to estimate the temperature profile is too complex for the current manuscript, but I am still convinced that it is a minor effort (given the complex simulations!) and would be highly beneficial and I highly recommend to consider this for FUTURE studies.

We agree that it is a logical next improvement of our method, especially if the purpose is retrieving (accurate) dry snowpack properties.

My point on the performance is obsolete - this was a misunderstanding on my side.

However, to the melt rate is indeed a very relevant quantity for this manuscript, as the (total or volumetric) water content is highly influenced by the amount of water available for infiltration/percolation into the snow. The data is provided by RCMs and can easily be analysed. Only comparing to ERA2m temperatures (given the atmospheric inversion in the 2m above the snow surface) seems to me not sufficient.

This refers to a comment of the first review that was not clear to us, as indicated in our response. We agree that the amount of liquid water present in the snowpack is controlled by the melt rate, at least its maximum value, while the minimum value is also controlled by the refreezing capacity of the snowpack (cold content) and other processes. However, we do not understand how this fits in the present analysis, where we don't try to predict the liquid water content, we "only" explore the sensitivity of the brightness temperature to prescribed amounts of liquid water.

Line 344: Banwell in preparation cannot be cited

We have removed the reference.